# Tissue-specific cell-free DNA degradation quantifies circulating tumor DNA burden

Guanhua Zhu[1], Yu A. Guo [1], Danliang Ho[1], Polly Poon[1], Zhong Wee Poh[1], Pui Mun Wong[1], Anna Gan[1], Mei Mei Chang[1], Dimitrios Kleftogiannis[1], Yi Ting Lau[1], Brenda Tay[2], Wan Jun Lim[2], Clarinda Chua[2], Tira J. Tan[2], Si-Lin Koo[2], Dawn Q. Chong[2], Yoon Sim Yap [2], Iain Tan[1,2,3✉], Sarah Ng[1✉] & Anders J. Skanderup [1,2✉]

Profiling of circulating tumor DNA (ctDNA) may offer a non-invasive approach to monitor disease progression. Here, we develop a quantitative method, exploiting local tissue-specific cell-free DNA (cfDNA) degradation patterns, that accurately estimates ctDNA burden independent of genomic aberrations. Nucleosome-dependent cfDNA degradation at promoters and first exon-intron junctions is strongly associated with differential transcriptional activity in tumors and blood. A quantitative model, based on just 6 regulatory regions, could accurately predict ctDNA levels in colorectal cancer patients. Strikingly, a model restricted to blood-specific regulatory regions could predict ctDNA levels across both colorectal and breast cancer patients. Using compact targeted sequencing (<25 kb) of predictive regions, we demonstrate how the approach could enable quantitative low-cost tracking of ctDNA dynamics and disease progression.

[1] Genome Institute of Singapore (GIS), A*STAR, Singapore, Singapore. [2] National Cancer Center Singapore, Singapore, Singapore. [3] Duke-NUS Medical School, National University of Singapore, Singapore, Singapore. ✉email: tanbhi@gis.a-star.edu.sg; ngbhs@gis.a-star.edu.sg; skanderupamj@gis.a-star.edu.sg

Cell-free DNA (cfDNA) is present in the blood circulation of humans. In healthy individuals, the death of normal cells of the hematopoietic lineage is the main contributor of plasma cfDNA[1]. In cancer patients, blood plasma can carry circulating tumor DNA (ctDNA) fragments originating from tumor cells, offering non-invasive access to somatic genetic alterations in tumors. The ctDNA profile of a cancer patient is clinically informative in at least two major ways. Firstly, the profile can provide information about specific actionable mutations that can guide therapy[2–5]. Secondly, the profile can be used to infer tumor growth dynamics by estimating the amount of ctDNA in the blood[6,7]. This latter information offers a promising non-invasive approach to track disease progression during clinical trials or therapy, offering a real-time tool to adjust therapy[8,9].

Existing next-generation sequencing-based approaches to estimate ctDNA levels in plasma samples are based on somatic single nucleotide variant allele frequencies (SNV VAFs), copy number aberrations (CNAs), or DNA methylation patterns[10,11]. cfDNA targeted sequencing typically only covers a few hundred selected cancer genes because of the need for ultra-deep sequencing (~10,000x). ctDNA burden estimation based on SNVs may therefore be challenging when no clonal mutations exist among the targeted genes. Alternatively, low-pass whole-genome sequencing (lp-WGS) yields segmental/arm-level CNAs that also allow for inference of ctDNA burden[12]. However, some cancers may not have sufficient levels of aneuploidy and chromosomal instability[13,14] needed for robust estimation. Sequencing of DNA methylation patterns may provide a general approach to quantify the cellular origin of cfDNA[15]. However, both DNA methylation and lp-WGS profiling require separate assays in addition to standard targeted gene sequencing, highlighting the need for approaches that simultaneously allow for profiling of actionable cancer mutations and quantitative estimation of ctDNA burden.

Previous studies have shown that the size distribution of cfDNA fragments has a mode of ~166 bp, suggesting that nucleosome-bound DNA fragments are preserved during cell death and shed into the circulation[16,17]. Nucleosome depleted regions (NDRs) are therefore more frequently degraded, yielding a nucleosome-dependent degradation footprint in cfDNA profiles, which can be used to infer tissue of origin[18–20]. Moreover, plasma cfDNA degradation patterns in cancer patients have been used to infer tumor gene expression[18,21]. Here, using these concepts, we hypothesized that a limited set of tumor or blood-specific NDRs could be used to infer the ctDNA burden (fraction) in the blood circulation of cancer patients. ctDNA burden refers to the relative amount of ctDNA out of all cfDNA molecules in a plasma sample. Using deep cfDNA WGS data from cancer patients and healthy individuals, we trained and tested a quantitative model that infers the ctDNA burden using cfDNA sequencing data from a limited set of NDRs. We show that this model is accurate for plasma samples from both colorectal cancer (CRC) and breast cancer (BRCA) patients (mean absolute error ≤4.3%), and we explore how it can be deployed using a compact targeted sequencing assay for low-cost and quantitative tracking of patient ctDNA dynamics.

## Results

**Overview of approach.** We collected blood samples ($n = 29$) from healthy individuals and extracted plasma cfDNA for paired-end WGS (merged ~150x coverage) (Fig. 1). We performed targeted sequencing (see Methods section) of plasma samples (CRC $n = 65$, BRCA $n = 36$) from cancer patients and selected samples (CRC $n = 12$, BRCA $n = 10$) with high SNV VAFs (indicating high ctDNA burden) for deep ~90x cfDNA WGS (Supplementary

Data 1). In these high ctDNA burden WGS samples, we could obtain ctDNA burden estimates using existing methods[22–25] that infer tumor purity using matched tumor and germline high-depth WGS data. To identify candidate NDR features for the quantitative model, we identified tumor and blood-specific genes with differential NDR cfDNA degradation in their promoters and first exon–intron junctions in plasma samples from healthy individuals and cancer patients. Using a machine learning and in silico cfDNA generation approach, we then trained and tested a sparse linear model to predict ctDNA burden from NDR cfDNA coverage. To further explore how the approach could be useful for cost-effective monitoring of ctDNA dynamics, we designed a compact (<25 kb) capture-based sequencing assay targeting predictive NDRs to explore the robustness of NDR-based targeted approach using independent plasma samples ($n = 53$) from CRC patients, and applied it to estimate ctDNA levels in longitudinally collected plasma samples from a cohort of five colorectal cancer patients.

**Association of gene expression and cfDNA fragmentation patterns.** Analysis of cfDNA from the healthy individuals expectedly revealed nucleosome depletion and reduced cfDNA protection flanked by a series of strongly positioned nucleosomes at gene promoter regions (Fig. 2a). Consistent with a previous study[21], relative coverage at the promoter NDR was inversely correlated with gene expression in whole blood cells. Studies of nucleosome positioning in cells have found that, apart from promoters, exon–intron junctions are associated with NDRs[26,27]. We therefore systematically scanned these gene regions for association between gene expression and cfDNA relative coverage (Fig. 2a). Strikingly, we found that the first exon–intron junction of transcripts showed a similar association between coverage and expression, where relative cfDNA coverage at the NDR ranging from −300 to −100 bp with respect to the end of the first exon exhibited a strong inverse correlation with transcript expression in whole blood cells. However, surprisingly, correlation between expression and cfDNA coverage was not observed at other exon–intron and intron–exon junctions as well as at gene ends (Fig. 2a and Supplementary Fig. 1). As expected, when comparing highly expressed (fpkm ≥ 30) and unexpressed gene groups, we observed a strong positive correlation (Pearson $r = 0.81$; Spearman correlation, $\rho = 0.85$) between the cfDNA relative coverage at promoter and first exon–intron junction NDRs across genes (Fig. 2b). While relative coverage at these NDRs correlated strongly with gene expression level, relative coverage could not perfectly separate unexpressed from expressed genes (Fig. 2b, c), suggesting that additional factors beyond gene expression contribute to NDR cfDNA degradation. To further explore the factors affecting cfDNA degradation at NDRs, we explored the association between NDR relative coverage and a range of epigenetic features (Supplementary Fig. 2). In addition to gene expression levels (linear regression, promoter $r = −0.23$, junction $r = −0.22$), relative coverage was negatively correlated with DNase hypersensitivity (promoter $r = −0.60$, junction $r = −0.55$), H3K4me3 (promoter $r = −0.59$, junction $r = −0.54$), and H3K27ac (promoter $r = −0.45$, junction $r = −0.41$), which are markers of open chromatin, active promoters, and active enhancers respectively[28]. In contrast, H3K36me3 (promoter $r = 0.49$, junction $r = 0.46$) and H3K9me3 (promoter $r = 0.11$, junction $r = 0.10$), markers of gene bodies and heterochromatin, were positively correlated with NDR relative coverage.

**cfDNA coverage patterns at NDRs in colorectal cancer patients.** To further explore the hypothesis that NDR cfDNA coverage in plasma samples from cancer patients is associated

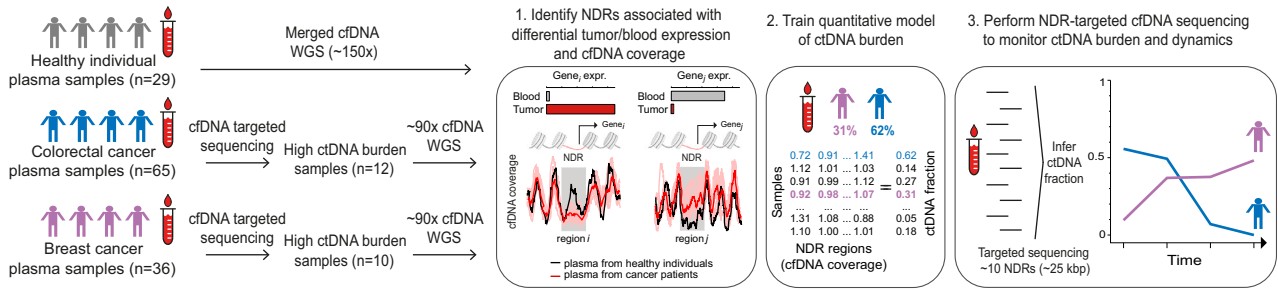

**Fig. 1 Overview of approach.** Deep cfDNA WGS profiles of plasma samples from healthy individuals and cancer patients were compared to identify nucleosome-depleted regions (NDRs) with tumor/blood tissue-specific expression and differential cfDNA coverage. A model was trained to predict ctDNA levels from NDR cfDNA coverage. A compact assay targeting predictive NDRs was used to perform longitudinal profiling of ctDNA levels and dynamics.

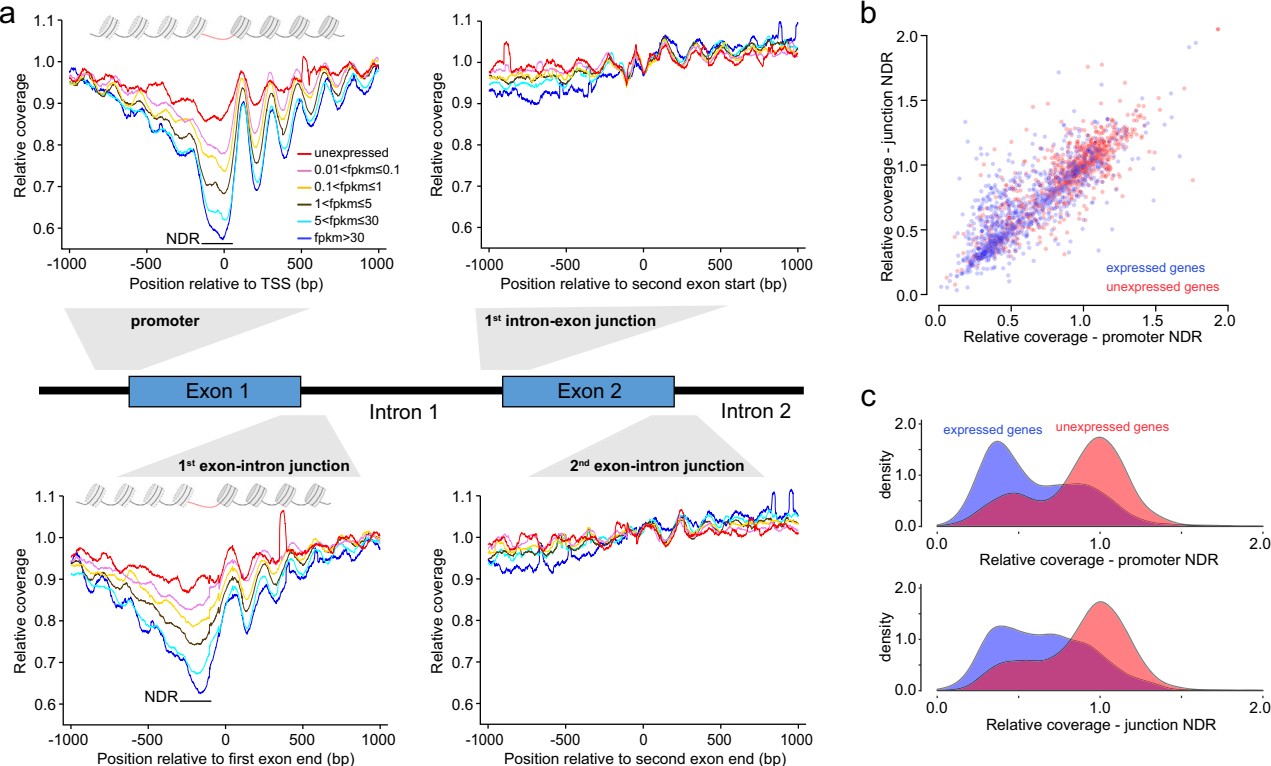

**Fig. 2 Characteristics of cfDNA degradation patterns at promoters and exon–intron junctions. a** Systematic analysis of gene regulatory regions for association of gene expression and cfDNA relative coverage. Relative coverage refers to cfDNA coverage across the given region when normalized to ±1 kb flanking regions. The nucleosome-depleted regions of promoter (NDR, −150 to 50 bp relative to TSS) and first exon–intron junction (NDR, −300 to −100 bp relative to first exon end) are highlighted. **b** Relative cfDNA coverage of promoter and junction NDRs for expressed (fpkm ≥30 in whole blood) and unexpressed genes. **c** Distribution of promoter and junction NDR relative coverage for expressed and unexpressed genes.

with the epigenetic state of tumor cells, we first used a targeted sequencing panel to screen plasma samples from CRC patients for cases of high ctDNA burden (VAF > 15% for known cancer driver mutations, Fig. 1). We initially identified 8 plasma samples from 5 patients and performed high-depth WGS on these samples (~72x–101x, Sample ID: CRC-1–8 in Supplementary Data 1). We inferred ctDNA fractions in these samples using four existing tissue-based estimation methods[22–25] (see Methods section) and used the median tumor purity estimate from these methods as ctDNA fractions (in the range 35–86%, Supplementary Data 1). We then used gene expression data from TCGA and GTEx to identify genes specifically expressed in CRC tumors and whole blood (see Methods section, Supplementary Fig. 3). As an example, we identified *PPP1R16A* as a CRC-specific gene with robust depletion of NDR cfDNA coverage in plasma samples from cancer patients as compared to healthy individuals, and

*GMFG* as a blood-specific gene with greater coverage depletion in healthy blood plasma (Fig. 3a). As expected, CRC-specific genes generally showed depletion of cfDNA at both promoter and junction NDRs in the plasma of CRC patients compared to healthy controls (Fig. 3b). In contrast, blood-specific genes showed higher cfDNA coverage at NDRs in the plasma of CRC patients compared to healthy controls. Furthermore, directly comparing CRC and blood-specific genes, CRC-specific genes had significantly greater cfDNA depletion at NDRs in plasma samples from CRC patients ($P < 2.2 \times 10^{-16}$, Wilcoxon rank-sum test, Fig. 3b).

**Quantitative estimation of colorectal cancer ctDNA burden.** With the insight that cfDNA coverage at NDRs is associated with the transcriptional state of DNA in the tumor cells, we hypothesized that cfDNA coverage at a small set of NDRs could be

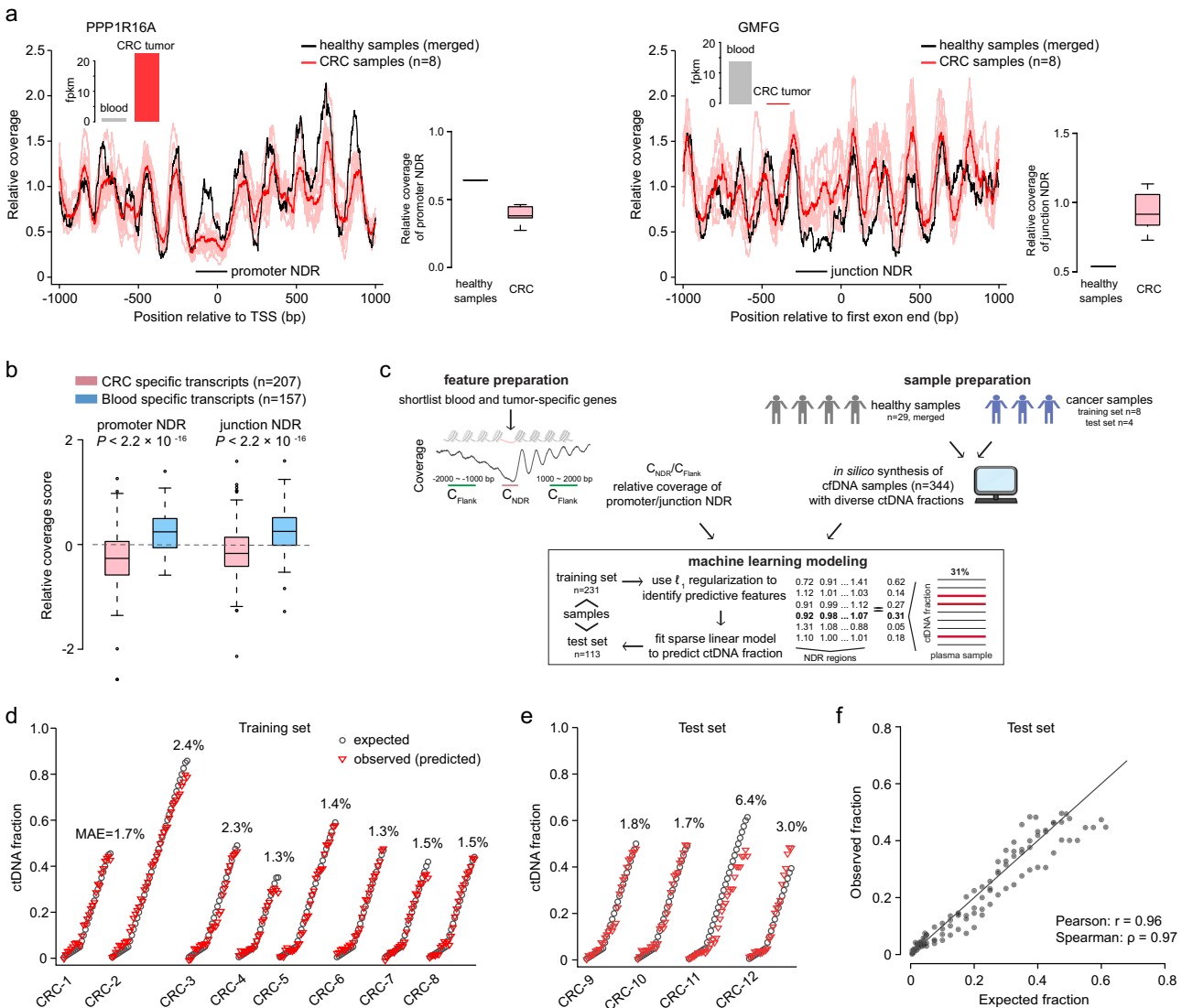

**Fig. 3 Quantitative estimation of colorectal cancer ctDNA burden. a** cfDNA relative coverage for the promoter region of *PPP1R16A* (ENST00000528430) overexpressed in CRC tumors relative to whole blood, and cfDNA relative coverage for the junction region of *GMFG* (ENST00000602185) overexpressed in whole blood relative to CRC tumors. The dark red curve shows the mean coverage across CRC samples. **b** Relative coverage score (see Methods) of NDRs in transcripts differentially expressed between CRC tumors and whole blood. Two-sided Wilcoxon rank-sum tests were performed to compare CRC and blood-specific transcripts. **c** Schematic showing how the predictive model of ctDNA fractions was developed: differentially expressed genes in CRC and blood were identified, NDR relative coverage features were obtained from in silico generated cfDNA samples, predictive features were selected, and a quantitative model was fitted. **d, e** Comparison of expected (in silico simulation) and observed ctDNA fractions across the CRC cfDNA samples in the **d** training and **e** test set, respectively. The mean absolute error (MAE) is listed for each sample. **f** Comparison between observed and expected ctDNA fractions of the 113 samples in the test set. Boxplots represent the median as centreline, the interquartile range (IQR) as bounds of box, and the lower quartile −1.5 IQR and the upper quartile +1.5 IQR as whiskers.

used to infer the ctDNA burden (fraction of tumor DNA out of all cfDNA) in the blood plasma of a cancer patient. As training data we in silico "diluted" 8 deep WGS samples from 5 CRC patients with data from healthy individuals, resulting in a training set of 231 samples of ctDNA proportions ranging from 0.5% up to the original undiluted fractions (Fig. 3c and Supplementary Table 1). We shortlisted candidate CRC-specific transcripts that were upregulated in CRC tumors ($\text{fpkm}_{\text{CRC}} > 10$, $\text{fpkm}_{\text{blood}} < 1$) and had a differential DNA degradation signal at both promoter and junction NDRs (relative coverage score < −0.2). Candidate blood-specific transcripts were shortlisted with similar criteria ($\text{fpkm}_{\text{CRC}} < 1$, $\text{fpkm}_{\text{blood}} > 10$, relative coverage score > 0.2). Relative coverages at the NDRs of these candidate transcripts

were used as input features (total 529 unique tumor and 379 blood features, Supplementary Data 2). We then used Lasso $L_1$-regularization regression in combination with a stability-based feature selection approach to a select a minimal set of 6 predictive NDRs (Table 1), which could predict the ctDNA fraction in the training data with a mean absolute error (MAE) of ~1.8% (Fig. 3d). Expectedly, the signs of coefficients for the 6 NDRs in the trained model corresponded to the sign of differential expression of the associated transcripts in tumor tissue relative to whole blood (Supplementary Table 2). To evaluate the ability of the model to generalize to unseen data, we sequenced 4 additional samples (CRC-9–12 in Supplementary Data 1, WGS at ~80–95x) from 2 new CRC patients and created an in silico diluted test set

**Table 1 NDR features predictive of ctDNA fraction in CRC.**

| Gene | Transcript | Chr | Site | Region | Expr. | FPKM$_{blood}$ | FPKM$_{CRC}$ | Pr. |
|------|-----------|-----|------|--------|-------|------------|----------|-----|
| *SHKBP1* | ENST00000599716 | 19 | 41,082,891 | Junction | Blood | 10.66 | 0.22 | 1.000 |
| *ACSL1* | ENST00000454703 | 4 | 185,747,070 | Junction | Blood | 35.07 | 0.78 | 1.000 |
| *BCAR1* | ENST00000162330 | 16 | 75,285,369 | Junction | Tumor | 0.00 | 16.86 | 1.000 |
| *RAB25* | ENST00000361084 | 1 | 156,030,951 | Promoter | Tumor | 0.07 | 131.50 | 0.999 |
| *PRTN3* | ENST00000234347 | 19 | 840,960 | Promoter | Blood | 13.78 | 0.00 | 0.995 |
| *LSR* | ENST00000605618 | 19 | 35,739,922 | Promoter | Tumor | 0.22 | 31.85 | 0.990 |

The column Site is the position of the nucleosome-depleted site (GRCh37); Region is the annotated class of the nucleosome-depleted site (promoter or exon-intron junction); Expr. denotes whether the transcript is specifically expressed in CRC tumor tissue or whole blood cells; Pr. is the probability/frequency with which the feature was selected in the Lasso stability-selection approach.

of 113 samples (Supplementary Table 1). The model accurately predicted the ctDNA proportion in this independent test set (Fig. 3e, MAE = 3.4%). A direct comparison shows high similarity between the observed (predicted) and expected ctDNA fractions (Fig. 3f; Pearson $r = 0.96$; Spearman correlation, $\rho = 0.97$). To further explore the performance of more complex models, we estimated the predictive error as a function of model complexity (number of top predictive features) and found that models with 4–10 NDR features were generally more accurate and better at generalizing to unseen data compared with models using fewer or more features (Supplementary Fig. 4). Next, we explored the lower limit for ctDNA detection in the NDR model. We evaluated the sensitivity and specificity of the model as a function of ctDNA fraction threshold. We used the 113 in silico test set CRC samples (Fig. 3e, CRC-9–12) as positives, and 40 random subsets (Supplementary Data 3, ~80x each) from the data of plasma samples ($n = 29$) from healthy individuals as negatives. At a 2% ctDNA fraction threshold, the model correctly predicted all positive (of expected fractions ≥2% threshold) and negative samples (100% sensitivity and specificity, Supplementary Data 3). In comparison, at a 1% threshold, the sensitivity was maintained at 100% but the specificity dropped to 75%.

To further evaluate the robustness of the model when tested on in silico samples generated using healthy samples not seen during model training, we split the healthy samples ($n = 29$) into two different groups to separately generate in silico training and test data. Reassuringly, this analysis showed robust model performance in the presence of independent train/test healthy samples (Supplementary Fig. 5; test set median Pearson $r = 0.92$; median Spearman correlation, $\rho = 0.93$; median MAE = 5.3%).

Next, we compared the predictive performance of our model with ichorCNA[12], a method that estimates the ctDNA fraction on the basis of arm-level copy number alterations in low-pass WGS data. Overall, ichorCNA accurately predicted ctDNA burden (Supplementary Fig. 6; Pearson $r = 0.91$; Spearman correlation, $\rho = 0.92$). However, while 31 out of 120 low burden samples (ctDNA burden ≤5%) were predicted as non-cancerous by ichorCNA (Supplementary Fig. 7), only 4/120 were predicted as non-cancerous by the NDR approach. This is consistent with the reported 3% lower limit of detection using arm-level CNAs[12].

**Targeted NDR assay to estimate ctDNA burden.** Intriguingly, since the predictive models used data from only a few NDRs, we hypothesized that a targeted sequencing approach could be deployed for robust and low-cost estimation of ctDNA burden. The CRC model only requires cfDNA relative coverages at 6 NDRs (Table 1). We therefore designed capture probes for these six regions (total ~24 kb) and performed targeted sequencing (~300x) on 53 new plasma samples from CRC patients (Fig. 4a), followed by ctDNA burden estimation from the relative coverage of the NDRs using the existing CRC model. To examine the

accuracy of our model on targeted NDR sequencing data, we also performed low-pass WGS (~4x) on the same plasma samples for ctDNA content estimation with the CNA-based method, ichorCNA. We observed high concordance (Pearson $r = 0.84$; Spearman correlation, $\rho = 0.79$) of estimated ctDNA burden with the CNA and NDR-based approaches (Fig. 4b and Supplementary Data 4). Moreover, in the 53 samples, we performed targeted sequencing (~6000x) of a panel of 100 frequently mutated genes (~370 kb, Supplementary Data 5) in colorectal cancer. SNVs called by MuTect and VarScan were intersected and further filtered to minimize false positives (Supplementary Data 6, see Methods section). This analysis identified high-confidence somatic mutations in 27 plasma samples and revealed high correlation (Pearson $r = 0.85$; Spearman correlation, $\rho = 0.88$) between maximum VAFs and NDR-based ctDNA burden estimates across samples (Fig. 4c and Supplementary Data 4). ctDNA was detected in 49 out of 53 (92%) samples with the targeted NDR approach, compared to 33/53 (62%) and 27/53 (51%) with ichorCNA and SNV calling approaches, respectively. The four ctDNA-negative samples identified with the NDR approach were also ctDNA-negative using ichorCNA and the SNV approach (Supplementary Data 4). Overall, this demonstrates that the NDR-based estimation approach is robust and can be deployed with a compact and low-cost targeted sequencing approach.

**NDR-based monitoring of ctDNA dynamics and disease progression.** To further explore how NDR-based ctDNA burden estimation could be used for low-cost monitoring of cancer progression, we applied the targeted NDR assay to serial plasma samples collected from five CRC patients (Fig. 4d). Overall, targeted NDR profiling showed concordant ctDNA burden dynamics when compared with SNV VAFs profiled in the same samples, with coinciding increases and decreases in ctDNA burden and VAFs over time. For example, patient C357 showed generally increasing ctDNA burden and VAFs over time, and patient C986 had an intermediate coinciding peak in both ctDNA burden and VAFs. We detected driver mutations in *TP53*, *PIK3CA* and *APC* in patient C986. While VAFs of these mutations were highly correlated, they showed a between-mutation spread of ~0.1–0.2 VAF units across all timepoints. Similarly, patient C519 had *TP53* and *APC* mutations with a ~0.2–0.3 unit difference in VAFs. While such differences may be caused by both technical (e.g. capture efficiency) and biological (e.g. clonality or concomitant CNAs) bias, they demonstrate the challenge in estimating ctDNA burden levels based on VAFs alone.

We noted a number of plasma samples for which the NDR-based ctDNA burden was inferred to be positive, yet our variant calling pipeline identified no SNVs under default settings. To further understand this discordance, we manually inspected the raw sequencing data in these "mutation-free" plasma samples. Indeed, when searching for variants that were identified in other

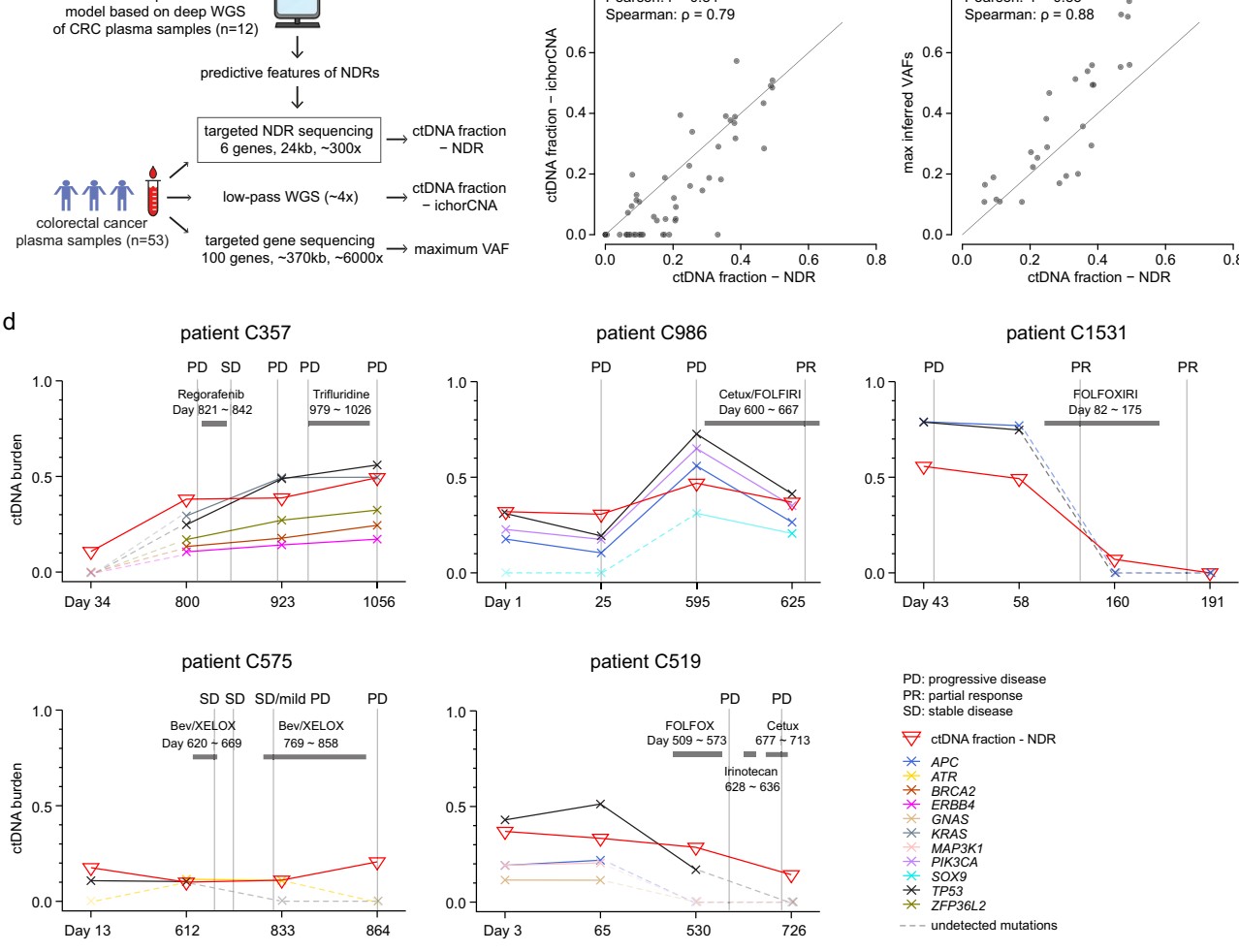

**Fig. 4 Targeted NDR assay to quantify ctDNA burden and monitor cancer progression. a** Schematic showing how targeted NDR sequencing, low-pass WGS, and targeted gene sequencing were performed on a cohort of 53 CRC plasma samples. **b** Comparison of ctDNA fractions ($n = 53$) inferred by targeted NDR sequencing and low-pass WGS (ichorCNA). **c** Comparison of ctDNA fractions ($n = 27$) inferred by targeted NDR sequencing and maximum VAFs (maximum VAF of all SNVs identified in a given plasma sample). **d** NDR-quantified ctDNA burden across serial plasma samples and its association with events of cancer progression and treatment response. Somatic SNV VAFs are highlighted for each timepoint; SNVs detected in at least two timepoints are shown. SNVs undetected with standard filtering criteria at given timepoints are indicated with a dashed line. Treatment types and intervals are highlighted. Events of disease progression as inferred by computerized tomography (CT) scans are shown.

samples/timepoints from the same patients, the raw sequencing data supported presence of the expected SNVs in all the samples with positive NDR-quantified ctDNA burden (Supplementary Data 7). In contrast, one plasma sample (patient C1531, day 191) was quantified with zero ctDNA burden by the NDR approach and manual screening confirmed absence of *TP53* and *APC* mutations in this sample (Supplementary Data 7). Overall, these results highlight the robustness of the targeted NDR assay for ctDNA burden estimation.

We next explored how ctDNA burden dynamics correlate with response to targeted or cytotoxic treatments. Patient C357 was treated with Regorafenib (days 821–842 after diagnosis) followed by Trifluridine (days 979–1026). However, ctDNA burden estimation in this time interval (days 800–1056) showed no drop in ctDNA burden following either treatment, indicating tumor resistance to both drugs; end-treatment CT scans (day 916 and 1056 respectively) confirmed progressive disease. In contrast, patients with positive response to treatment showed a marked reduction of ctDNA burden in plasma. For example, patient C1531 received the chemotherapy regimen of FOLFOXIRI (days 82–175) and had on and post-treatment CT scans showing partial

response. Strikingly, this patient showed a concomitant and marked drop in ctDNA burden both during (day 160) and after (day 191) treatment. In patient C575, *TP53* and *ATR* mutations were only identified at two out of four timepoints by our pipeline. In this patient, both CT scans and ctDNA burden estimation inferred stable disease (days 612–833) during the first round of XELOX/bevacizumab treatment. However, during the second round of treatment, the ctDNA burden increased (day 864) and CT scans confirmed progressive disease, indicating acquired drug resistance. Finally, discrepancies between tumor dynamics inferred from CT imaging and ctDNA burden have previously been reported[6,29]. Patient C519 reflected one such example, where CT scans indicated progressive disease while both ctDNA burden estimates and mutation VAFs decreased.

**Estimation of ctDNA burden across cancer types**. The predictive model for CRC ctDNA burden included 3 (out of 6) NDR coverage features from genes overexpressed in whole blood. Intriguingly, a predictive model completely restricted to blood-specific genes could hypothetically quantify the extent that a cfDNA profile deviates from a healthy baseline profile, allowing

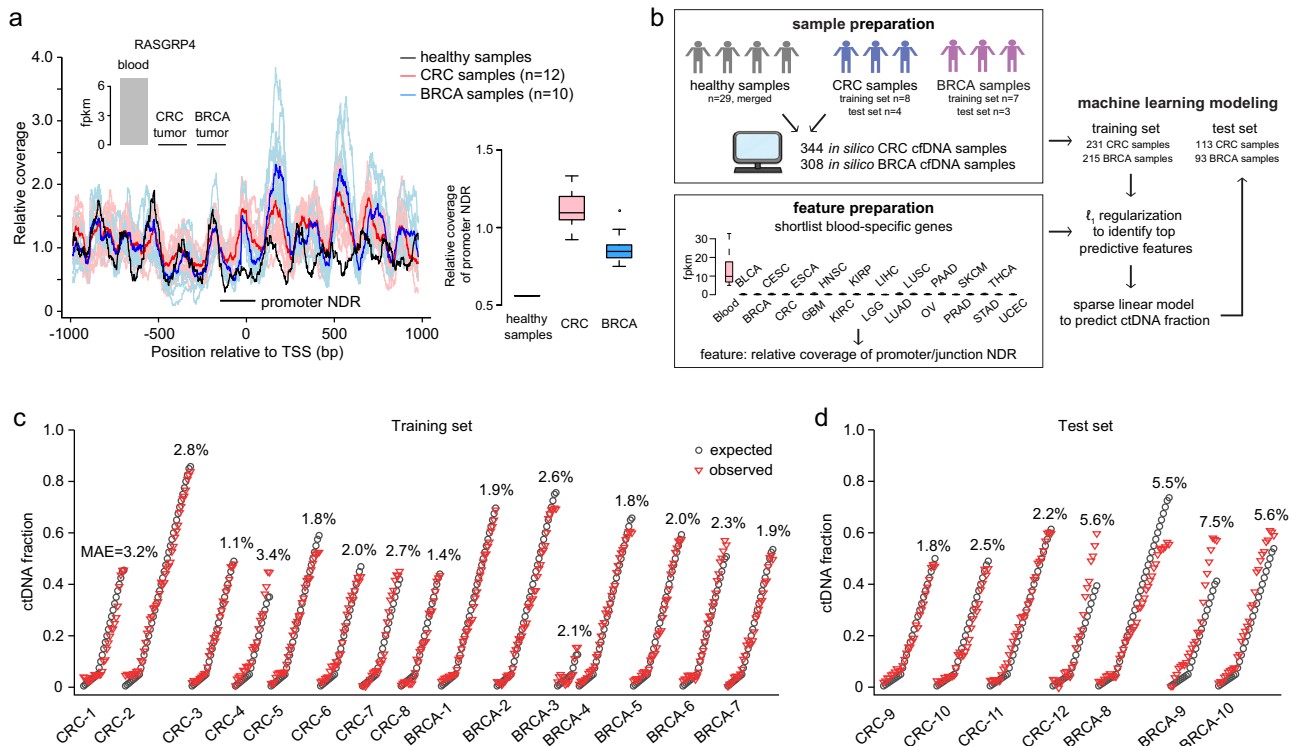

**Fig. 5 Estimation of ctDNA burden across two distinct cancer types. a** cfDNA relative coverage across the promoter region of the blood-specific gene, *RASGRP4* (ENST00000615340). Red and blue curves show the mean of the coverages from plasma samples from CRC and BRCA patients, respectively. **b** Schematic showing how the ctDNA content prediction model across CRC and BRCA cfDNA samples was developed. **c, d** Comparison of expected (in silico simulation) and observed ctDNA fractions across CRC and BRCA cfDNA samples in the **c** training and **d** test set, respectively. The mean absolute error is listed for each sample. Boxplots represent the median as centreline, the interquartile range (IQR) as bounds of box, and the lower quartile –1.5 IQR and the upper quartile +1.5 IQR as whiskers.

prediction of ctDNA burden across different cancer types. Indeed, we were able to identify genes overexpressed in whole blood compared to solid tumor tissue that also had decreased NDR coverage in plasma samples from healthy individuals as compared to patients of distinct cancer types (Fig. 5a). To further systematically test this idea, we performed deep (~72–94x) cfDNA WGS sequencing of blood samples from 10 breast cancer patients (Supplementary Data 1) and generated in silico diluted cfDNA samples of variable ctDNA burden (Supplementary Table 1 and Fig. 5b). Since germline data was not available for the BRCA patients, ctDNA fractions were estimated in the 10 BRCA samples using ichorCNA. We then shortlisted transcripts highly overexpressed in whole blood compared to solid tumor tissues (comprising 20 different solid tumor types, Fig. 5b), yielding 792 blood-specific candidate NDR features (Supplementary Data 8). Using a training dataset comprising cfDNA samples from both CRC and BRCA patients, we found that models comprising approximately 10 features were able to generalize well to unseen data from both cancer types (Supplementary Fig. 8). A model fitted with the training data using the top 10 predictive features (Supplementary Data 9) had a mean absolute error of 2.2%, with comparable accuracy in CRC and BRCA samples (Fig. 5c). In the unseen CRC and BRCA test data (Fig. 5d), the model achieved an overall accuracy (MAE = 4.3%; Pearson $r = 0.95$; Spearman correlation, $\rho = 0.97$; Supplementary Fig. 9), comparable to the CRC-specific model applied to CRC data (MAE = 3.4%; Pearson $r = 0.96$; Spearman correlation, $\rho = 0.97$; Fig. 3e, f) and a BRCA-specific model applied to the BRCA data (MAE = 6.1%; Pearson $r = 0.97$; Spearman correlation, $\rho = 0.97$; Supplementary Fig. 10). We also observed strong concordance between the CRC + BRCA and CRC-specific models in their predicted ctDNA fractions in

the test set plasma samples from CRC patients (Pearson $r = 0.95$; Spearman correlation, $\rho = 0.95$; Supplementary Fig. 11). We analyzed the lower limit of detection for the CRC + BRCA model by evaluating the sensitivity and specificity of the model as a function of ctDNA fraction threshold. We used the 206 in silico test set samples (113 CRC + 93 BRCA, Fig. 5d) as positives and 40 random subsets (~80x each) from healthy individuals as negatives. At a 3% ctDNA fraction detection limit, the CRC + BRCA model achieved 100% sensitivity and specificity (Supplementary Data 10). In comparison, at a 2% threshold, the sensitivity was almost maintained at 99.5% but the specificity dropped to 88%. These results support that a model based on a limited set of blood-specific NDR features can predict ctDNA fractions across two distinct cancer types.

## Discussion

Monitoring of ctDNA offers a non-invasive approach to tracking disease progression and has been demonstrated as a valuable real-time tool for assessing therapeutic response[3,30–34]. Here, we show that cfDNA coverage patterns at tumor and blood-specific NDRs can be used for quantitative estimation of the ctDNA burden in blood plasma samples. While SNV VAFs can be used as a proxy for the ctDNA burden[35], this only works for the subset of patients with known and measured clonal SNVs in a given targeted gene panel. SNV-based approximation of ctDNA burden may be further challenged by clonal haematopoiesis[36], which is frequently observed in cancer patients. Additionally, absolute ctDNA fraction estimation from SNVs requires co-estimation of allele zygosity and clonality[37], which may be challenging to infer for metastatic patients with multiple independently evolving tumors contributing ctDNA to the blood circulation[38]. Furthermore, in

low ctDNA burden samples, which are common and clinically important[39], NDR-based burden estimation showed improved accuracy as compared to a lp-WGS-based estimation method[12]. In contrast to lp-WGS and DNA methylation-based profiling, NDR-based estimation is directly compatible with targeted gene panel sequencing. Since the ctDNA burden estimation model requires data from 10 or less NDRs, these regions can be profiled at low cost by capturing <25 kb of genomic sequence. Targeted cfDNA assays often cover hundreds of genes and >1 Mb captured genomic sequence, with larger panels required for profiling across cancer types and tumor mutation burden estimation[40]. It would be straightforward to co-profile NDRs in such assays, with only a minor increment in panel size. Furthermore, down-sampling analysis showed that the NDR approach is robust down to 100x sequence coverage (Supplementary Fig. 12), imposing a sequencing demand equivalent to ~0.001x WGS, orders of magnitude lower than current lp-WGS approaches. Importantly, an integrated NDR/gene assay would be able to estimate ctDNA burden in patients without clonal mutations in targeted cancer genes, potentially corresponding to 5–70% of patients depending on cancer type[41]. While the estimated lower limit of detection (~2%) of the NDR approach is likely not suited for screening of cancer in healthy/cancer-free individuals, the approach could enable low-cost and simultaneous quantitative estimation of ctDNA burden and mutational profiling in response to treatment interventions. Furthermore, critical for treatment decision support, independent ctDNA burden estimates could assist in classification of clonal and subclonal actionable mutations. Intriguingly, we found that a model restricted to blood-specific NDRs could robustly predict ctDNA burden across both colorectal and breast cancer patients, suggesting it might be possible to estimate ctDNA burden independently of tumor types and metastatic lesions. However, studies across multiple cancer types are needed to further test and establish the robustness of such an approach. We also recognize that the targeted NDR sequencing approach should be tested across larger patient cohorts and healthy individuals to more accurately assess potential technical bias and its limit of detection.

Nucleosome positioning across gene bodies, and its association with transcriptional activity, has been studied using both biochemical assays[27] and cfDNA profiles[18]. Unexpectedly, our systematic analysis across ordered exon–intron junctions revealed that, in addition to the promoter, only the first exon–intron junction showed signatures of strong nucleosome and expression-dependent cfDNA degradation (Fig. 2a and Supplementary Fig. 1). Interestingly, transcription and splicing are coupled processes[42], and it has been observed that H3K4me3 and H3K9ac chromatin marks of active transcription are concentrated specifically at both promoters and ends of first exons[43]. Our data further supports that nucleosome depletion and DNA accessibility at the first exon–intron splice junction is strongly associated with transcriptional activity, supporting a model where the first exon splice region may act as a transcriptional enhancer[43,44].

In summary, we show how tissue and expression-specific cfDNA degradation at NDRs can be used to quantitatively estimate ctDNA burden in blood samples. The approach is directly compatible with targeted gene sequencing, allowing for low-cost and simultaneous discovery of actionable cancer mutations and accurate estimation of ctDNA burden. We anticipate these insights will be useful in the design of next-generation cfDNA assays to quantitatively track and analyze cancer disease progression across time and patients.

## Methods
**Plasma samples**. Cancer patient and healthy volunteer samples were collected under studies 2013/110/B (now 2018/2795), 2013/251/B, 2014/119/B, and 2012/733/B

approved by the Singhealth Centralised Institutional Review Board. The written informed consent was obtained from the patients. Information of patients is provided in Supplementary Data 11. Plasma was separated from blood within 2 h of venipuncture via centrifugation at 10 min × 300 g and 10 min × 9730 g, and then stored at −80 °C. DNA was extracted from plasma using the QIAamp Circulating Nucleic Acid Kit following manufacturer's instructions. Sequencing libraries were made using the KAPA HyperPrep kit (Kapa Biosystems, now Roche) following manufacturer's instructions and paired-end sequenced (2 × 151 bp) on either an Illumina Hiseq4000 or HiseqX.

**Whole-genome sequencing**. We first used a targeted sequencing panel (Supplementary Data 5) to screen plasma samples from CRC patients and selected 12 samples (Supplementary Data 1) of likely high ctDNA burden, having maximum VAF >15% for known CRC cancer driver mutations (Supplementary Data 12). Similarly, we selected 10 BRCA plasma samples of high ctDNA burden, with either VAF >15% based on a panel of 77 genes (Supplementary Data 13) of common breast cancer mutations (Supplementary Data 14), or alternatively, significant proportions (>20%) of short (length <150 bp) cfDNA fragments (Supplementary Data 1). It has been reported that short cfDNA fragments below 150 bp are enriched in high-ctDNA plasma samples[45]. Deep WGS (~90x) was performed on the 12 cfDNA samples from 7 CRC patients and 10 cfDNA samples from 10 BRCA patients (Supplementary Data 1). For the 5 CRC patients with 2 samples each, there was at least a 12 months interval between the two samples. We used bwa-mem[46] to align the WGS reads from healthy ($n = 29$, ~5x coverage), cancer (CRC $n = 12$, BRCA $n = 10$, ~90x coverage), and matched germline samples (CRC $n = 12$ ~30x coverage, not available for BRCA) to the hg19 human genome. Duplicates were marked using biobambam[47]. A previous study found that trimming reads from both ends increased the coverage signal of nucleosome positioning[21]. Similarly, we used BamUtil[48] to trim the original reads (~151 bp) from the two ends and preserved the central 61 bp to amplify the nucleosome-associated DNA degradation signal. BAM files of healthy individuals were merged using SAMtools merge function[49]. Low-pass WGS (~4x) was performed on 53 cfDNA samples from 23 CRC patients (Supplementary Data 4).

**Sample preparation for targeted sequencing**. Plasma and patient-matched buffy coat samples were isolated from whole blood within six hours from collection and stored at −80 °C. DNA was extracted with the QIAamp Circulating Nucleic Acid Kit, followed by library preparation using the KAPA HyperPrep kit. All libraries were tagged with custom dual indexes containing a random 8-mer unique molecular identifier. Targeted capture was performed on xGen custom panels (Integrated DNA Technologies) relevant to the experiment: (a) panel of 100 genes selected based on literature review for relevance to colorectal, see Supplementary Data 5 or (b) capture probes (Supplementary Data 15) targeting genomic regions (4 kb centered at the sites in Table 1) related to the 6 NDRs predictive of ctDNA content in colorectal cancer. Paired-end sequencing (2 × 151 bp) was done on an Illumina Hiseq4000 machine. Information of the samples with NDR-positive ctDNA detection but SNV and ichorCNA-negative has been provided in Supplementary Data 16.

**Variant calling and allele frequency estimation**. Sequencing data was analyzed using the bcbio-nextgen pipeline[50], including read alignment with BWA mem, PCR duplicate marking with biobambam, as well as recalibration and realignment with GATK[51]. Somatic variant calling was performed using MuTect[52] and VarScan[53] with default parameters, and all calls were annotated with Variant Effect Predictor[54]. Variants were removed if they were outside coding regions. The inferred VAFs were either from one of the two callers if the variant was missed by one caller, or the mean if the variant was called by both callers (Supplementary Data 6). Variants from HLA-A, KMT2C and MUC17 were filtered because the majority of variants in these genes were also found by at least one caller at ≥0.005 VAF in buffy coat sequencing.

**Gene expression analysis**. Tissue-specific RNA-seq transcript expression data based on GTEx dataset (including 337 whole blood samples) and tumor RNA-seq transcript expression based on TCGA dataset were obtained from UCSC Toil RNAseq Recompute Compendium (Supplementary Table 3)[55]. Because a gene usually comprises multiple alternative transcripts with different genomic positions, we studied gene expression at the transcript level for a precise mapping of promoter and junction locations. Transcripts of all coding genes were grouped on the basis of their expression level (fpkm) in whole blood. If a group (e.g. 0.1 < fpkm ≤ 1; 25,155 transcripts) had more than 5000 transcripts, we randomly selected 5000 transcripts to represent the group. Unexpressed genes were defined as transcripts that were not expressed in ≥99% of all 7861 GTEx samples.

**Relative cfDNA coverage estimation**. Read coverage at promoter and junction regions was computed from BAM files with SAMtools depth function[49]. For the promoter region (−150 to 50 bp relative to TSS), the mean raw coverage across the region was divided (yielding "relative coverage") by the mean coverage of the upstream (−2000 to −1000 bp relative to TSS) and downstream (1000–2000 bp relative to TSS) flanks (Supplementary Fig. 13). Thus, the mean coverage of the

combined upstream and downstream 2k bp flanks serves as a "normalization factor". A similar approach was used for exon–intron junctions. To measure the difference of relative coverage at NDRs between plasma samples from CRC patients and healthy individuals, we computed the relative coverage score:

$$score = \frac{mean(CRC) - mean(healthy)}{\sqrt{var(CRC)}} \qquad (1)$$

where mean(CRC) and mean(healthy) are the mean of average relative coverages at NDRs across CRC plasma and healthy plasma samples respectively, and var(CRC) is the standard deviation of average relative coverages at NDRs across CRC samples. The variance across individual sites in healthy samples could not be estimated due to low sequencing depth (~5x). Instead, to test for differences in variance between cancer and healthy samples, we approximated the variance in healthy samples using bootstrapping. Briefly, we estimated the standard deviation across 20 subsets of healthy samples (~50x merged) generated by randomly sampling (with replacement) and merging 10 samples out of 29 healthy controls. This analysis showed similar separation between CRC and blood-specific NDRs when variance was estimated in plasma samples of cancer (Fig. 3b) and healthy control (Supplementary Fig. 14). When computing average relative coverage of each NDR (either −150 to 50 bp relative to TSS, or −300 to −100 bp relative to first exon end), positions with relative coverage >2 were truncated to reduce bias from potential outlier values.

To explore the association between relative coverage and a range of epigenetic features, we used linear regression to fit each candidate feature (covariate) with relative coverage (response). Whole blood gene expression (fpkm) was discretized into 6 bins [unexpressed, $0.01 < fpkm \leq 0.1$, $0.1 < fpkm \leq 1$, $1 < fpkm \leq 5$, $5 < fpkm \leq 30$, $fpkm > 30$] and fitted as a categorical covariate with the unexpressed group as the reference group. Peaks of epigenetic features [DNase, H3K4me3, H3K36me3, H3K27ac, H3K4me1, H3K9me3, and H3K27me3] from primary T-cells (E034) were obtained from the Roadmap Epigenomics Project. Epigenetic features were fitted as binary covariates with no signal as the reference group.

**Estimation of ctDNA fractions from deep WGS cfDNA data.** The ctDNA fractions in CRC plasma samples were quantified using four different methods: THetA2[22], TitanCNA[23], AbsCN-seq[24], and PurBayes[25]. These methods were originally developed to use matched tumor tissue and germline Exome/WGS data to estimate mutation and copy number tumor heterogeneity, including tumor purity. Here we applied these methods to the ~90x cfDNA and ~30x matched germline (buffy coat) WGS data to estimate ctDNA fractions. Somatic mutations and copy number alterations, as input to AbsCN-seq and PurBayes, were called by SMuRF[56] and CNVkit[57], respectively, using the bcbio-nextgen workflow[50]. The median of these four ctDNA fraction estimates for a given sample was used as the final consensus estimate of the ctDNA fraction. Since germline samples were not available for the BRCA patients, the ctDNA fractions of the BRCA plasma samples were estimated by ichorCNA[12].

**In silico sample generation.** The cancer cfDNA samples were in silico diluted by mixing cancer cfDNA reads with reads from healthy samples, maintaining the same average coverage as the original undiluted cancer cfDNA sample. The in silico generated samples were diluted from ctDNA content ranging from 0.005 up to the original undiluted fractions, with a denser sampling of low fractions ≤0.05 (Supplementary Table 1). We generated a training set of 231 samples originating from 8 samples from 5 CRC patients, and a test set of 113 samples originating from 4 samples from 2 additional CRC patients. For BRCA, the training set comprised 215 in silico generated samples from 7 patients/samples, and the test set had 93 samples from 3 patients/samples (Supplementary Table 1).

**Generation of NDR features.** We computed the relative coverage score (see above) of NDRs for all transcripts and combined the relative coverage score with expression data to shortlist tumor/blood-specific transcripts associated with differential tumor/blood NDR cfDNA coverage. For each transcript, we calculated its median fpkm ($fpkm_{blood}$) across all whole blood samples, its median fpkm ($fpkm_{CRC}$) across all CRC samples, as well as its respective median fpkm values for other tumor types. Tumor transcripts were defined as being highly expressed in CRC tumor, lowly expressed in normal blood cell, and more highly degraded in CRC samples at both promoter and junction NDRs ($fpkm_{CRC} > 10$, $fpkm_{blood} < 1$, relative coverage score $< -0.2$). Blood transcripts were defined with similar rules ($fpkm_{CRC} < 1$, $fpkm_{blood} > 10$, relative coverage score $> 0.2$). This approach shortlisted 284 CRC and 210 blood transcripts, each transcript with two features (promoter and junction NDR coverage). After removing overlapping features (multiple transcripts sharing the same NDR), we used NDR coverages of the resulting 529 tumor and 379 blood transcripts (total n=908) as input features for predictive modeling. For the CRC + BRCA model, we only shortlisted transcripts with blood-specific expression ($fpkm_{blood} > 5$) that were also lowly expressed ($fpkm < 1$) in tumors of all 20 tumor types (TCGA tumor type acronyms: BLCA, BRCA, CESC, CRC, ESCA, GBM, HNSC, KIRC, KIRP, LGG, LIHC, LUAD, LUSC, OV, PAAD, PRAD, SKCM, STAD, THCA, UCEC), leading to a total of 792 features.

**Lasso regularized regression to predict ctDNA fraction.** We used Lasso regularized linear regression using glmnet[58] to select features and predict ctDNA content in plasma cfDNA samples. To select robust features, we first extracted half of the training data randomly and used Lasso with ten-fold cross-validation to identify features predictive of ctDNA fractions. This procedure was repeated 1000 times and the top stable features (selection frequency ≥ 0.99) were extracted as the final predictive features, which resulted in 6 predictive features (Table 1) for the CRC-specific model and 10 predictive features (Supplementary Data 9) for the CRC + BRCA model, respectively. We trained the final predictive model with ten-fold cross-validation on the full training set. We also attempted to predict ctDNA fractions with log-transformed relative coverage, and tested the performance using a logistic regression model, both of which failed to outperform the current model in prediction accuracy (data not shown).

To evaluate the robustness of the model when we trained and tested on in silico samples generated using independent healthy samples, we split the normal samples evenly into 2 sets. The first set (N1) was used to perform in silico spike-ins/dilution of the training set, and the second set (N2) was used for in silico dilution of the test set. Briefly, we re-fitted the coefficients of the CRC model (comprising the 6 features in Table 1) using the training data (diluted with the N1 healthy samples), and we then evaluated the model accuracy on the withheld test samples (diluted with N2). We repeated this procedure 10 times and evaluated the model accuracy on the test data generated using the independent normal samples.

**ichorCNA benchmarking.** For the in silico samples, we mixed cfDNA reads from the 12 deep-WGS CRC samples with reads from healthy samples to generate in silico low-pass samples (~0.1x) for ctDNA content estimation using ichorCNA. We followed the usage guidelines with default parameters in the 2 step workflow: (1) read count coverage calculation with HMMcopy Suite, and (2) tumor content estimation with ichorCNA R package.

**Reporting summary.** Further information on research design is available in the Nature Research Reporting Summary linked to this article.

## Data availability

cfDNA sequencing data have been deposited at the European Genome-phenome Archive (EGA) under the accession code EGAS00001004657. Data is available under restricted access and will be released subject to a data transfer agreement. Tissue-specific RNA-seq transcript expression data (https://toil.xenahubs.net/download/gtex_RSEM_isoform_fpkm.gz) based on GTEx dataset and tumor RNA-seq transcript expression data (https://toil.xenahubs.net/download/tcga_RSEM_isoform_fpkm.gz) based on TCGA dataset were obtained from UCSC Toil RNAseq Recompute Compendium. The remaining data are available within the Article, Supplementary Information or available from the author upon request.

## Code availability

The code for generating coverage features and developing quantitative models is included in Supplementary Software 1. The NDR models, code, and data accessions are available at https://github.com/skandlab/NDRquant.

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

## Acknowledgements

This work was supported by A*STAR under its IAF-PP program (grant ID: H1801a0019).

## Author contributions

A.J.S., S.N., I.T., and G.Z. designed the study. P.M.W., B.T., W.J.L., C.C., T.J.T., S.L.K., D.Q.C., Y.S.Y., and I.T. provided samples and clinical information. P.P., A.G., Y.T.L., and S.N. performed the experiments. D.H. and D.K. performed mutation calling. A.J.S., G.Z., Y.A.G., Z.W.P., and M.M.C. analyzed the data. A.J.S. and G.Z. wrote the manuscript, with contributions from all authors.

## Competing interests

The authors declare no competing interests.
