## [Peer Review File · Nature Communications]

REVIEWER COMMENTS

Reviewer #1 (Remarks to the Author):

This is a fascinating study that shows that ctDNA burden can be directly predicted (and used as a surrogate for overall tumour burden) using a highly targeted sequencing panel. The study is generally extremely well-conducted, and the work will have very substantial impact on the field. There are three technical issues that should be addressed to strengthen or better support the conclusions that are drawn, but overall I would congratulate the authors on a really nice study!

1. Spike-Ins

- * spiking in a different individual (in silico dilution) has major challenges (SNPs, library prep, etc.)
- * it appears that the same normals were used in both the training & testing datasets, which leads to contamination of training & testing datasets
- * unclear exactly how "relative coverage" was defined
- * the validation set is going to require new in silico spike-ins with normal samples not used in the original study
- * please make the final coefficients for the predictive models available in Tables 1 and S5 (or elsewhere as appropriate -- every model needs its coefficients reported for reproducibility)

2. Data deposition

- * A significant fraction of the value in this study derives from the data it generates. This must be made available, with appropriate meta-data, in a suitable public repository.

3. NDR cfDNA degradation

- * The authors nicely show that expression is one correlate of this, but do not explore other possibilities. The authors need to do a systematic evaluation of other factors as best as possible. They may find nothing, but it is not appropriate, given the scope of this work, to leave that entirely to future studies.

Minor

- * in silico should not be hyphenated
- * please be clear about exactly how all deep cfDNA WGS samples were selected?
- * Page 3, line 6: the claim is made that only clonal mutations can be detected. No data is given to justify the idea that subclonal mutations are undetectable by ctDNA, and this is highly unlikely to be universally true
- * please use bp (not b) for genomic distances throughout
- * please do not use red & green together in the same plot

Reviewer #2 (Remarks to the Author):

The manuscript by Zhu et al. describes a novel method for quantifying plasma ctDNA burden in advanced cancer patients using a small set of tumour/blood-specific nucleosome depleted regions

(NDRs). The study takes advantage of the rationale reported previously (Snyder et al. Cell 2016; Ulz et al. Nat Genet 2016), demonstrating the relationship between cfDNA coverage at NDRs and the expression levels in the tumour/blood cells.

In addition to the assessment of cfDNA coverage in promoter regions, a further development represents the exploration of sequence coverage across gene bodies, specifically at exon-intron junctions, and its translation into designing a small sequencing panel to monitor disease progression in CRC patients.

The idea of designing a capture-based sequencing assay targeting tissue-specific NDRs is novel and represents a valuable, and potentially clinically relevant development. The authors first build a “CRC model” for estimating ctDNA burden from the relative coverage of selected tumour/blood-specific NDRs. They then exemplify its diagnostic value in a cohort of 53 advanced CRC patients (with average VAF = ~30%), in addition to applying the panel to longitudinally collected plasma samples of 5 CRC patients.

However, to fully demonstrate its analytical performance and clinical utility, some aspects of the study should be clarified. The manuscript might benefit from implementing suggested comments:

First, the study includes two sets of data for CRC and BRCA patients, but in fact only “CRC model” has been fully investigated. The development of “BRCA model” was not reported and, as a result, did not translate into designing a panel assay. It should be clarified whether there were any limitations in identification of “tumour-specific NDRs” in BRCA dataset that could potentially limit the application of NDR-based approach in other cancer types.

Instead, the authors built a “CRC+BRCA model” limited to only “blood-specific” NDRs, however without more details on different tumour types used for building this model (referred to as 20 different solid tumour types), and also without demonstrating its clinical use by designing a targeted panel assay. It is not clear whether some of the blood-specific NDRs identified in “CRC+BRCA model” overlap with the previous CRC model. This section of the manuscript (“CRC+BRCA model”) brings a confusion in understanding the flow of the document, since in the following section only the CRC model is applied. A suggestion would be to move this part at the end of the Result section. Regardless, developing “a universal NDR approach” across different cancer types using the presented “blood-specific model”, if applied across a variety of cancer types, would have had a much higher impact in the field in comparison with being limited to one cancer type. Justifying the selection of a particular design should be provided.

Second, the authors report on improved detection in “low ctDNA burden” cases using NDR-based approach, however, without addressing the levels of sensitivity/limit of detection. First, the terms “high ctDNA burden” and “low ctDNA burden” should be clarified. ctDNA levels in advanced cancer patients can be high based on SNV analysis, but in case these tumours are not driven by SCNA, ichorCNA would identify them as “low ctDNA burden”. The differences should be clearly defined in the manuscript.

This is in particularly relevant in the section “Quantitative estimation of colorectal cancer ctDNA

burden”, where the cases were pre-selected (and in silico diluted) based on the VAFs by SNV analysis, without considering their SCNA profiles. Therefore, the lower detection rates by ichorCNA could have possibly been biased just by the fact that these samples had low extent of SCNA (while still having high ctDNA levels identified by SNV analysis, Fig. S4). Can the authors add ichorCNA-based ctDNA fractions for CRC cases in Table S1 to assess this?

The authors report on a quantitative nature of their methodology, and thus should demonstrate the performance validation of the capture-based assay, and determine the levels of sensitivity. Although in silico dilutions of a limited number of WGS samples were prepared while building the model, the lower limits of ctDNA detection were not reported for the actual capture-based sequencing assay. The authors highlighted the clinical application of their method in a cohort of CRC patients, however to address the specificity of the capture-based assay, the approach would further require validation in a cohort healthy individuals.

The authors suggest to simultaneously profile actionable cancer mutations and quantitatively estimate ctDNA burden. To strengthen the message of the manuscript, the authors should demonstrate the actual benefit of NDR-based method over the other methods. To address this, it would be informative to determine the proportion of samples (among 53 CRC patients) detected by NDR-based approach while being SNV-negative, and conversely, to acknowledge the proportion of NDR-negative samples detected by SNV-based analysis. In particular, it would be important to highlight the samples detected by NDR-based analysis that were ichorCNA-negative (which appeared to have no SNVs detected). This section would benefit from providing a comprehensive supplementary data for all 53 patients showing their ctDNA levels based on ichorCNA, alongside the SNV analysis and the NDR-based analysis.

The Methods section provides a lack of information on explaining the prediction of ctDNA fractions using NDR-based method. Please add more details. Similarly, a clarification is needed in relation to estimation of the “relative cfDNA coverage score”. Was the relative coverage calculated within -1K to +1K region from TSS or specifically in the regions -150 to 50bp relative to TSS, or -300 to -100bp relative to first exon end?

The authors highlight that their method is cost-effective but this statement lacks a comparison with other methods. Additionally, could the authors perform in silico experiment to assess the effect of reducing the amount of sequencing on diagnostic performance of NDR-based analysis? At the moment, a coverage 300x is required but could this be further reduced to keep a similar performance?

Additional points:

- The authors mentioned that they performed targeted sequencing in breast cancer patients (page 13, line 5), but the data does not seem to be provided. Please clarify this. Additionally, please indicate specific locations corresponding to each gene in the panel (Table S7).
- In Table S6, please indicate that “coverage” corresponds to the coverage of sWGS data. In the main text, this table refers to “targeted sequencing assay” with a coverage of ~300x. Please correct this.
- Can the authors show correlations between ctDNA levels determined by “CRC-model” and

“CRC+BRCA” model in CRC patients? What is the concordance between predicted ctDNA fractions by the two separate models?

General response to reviewers

We thank the reviewers for their feedback and constructive comments, which have significantly improved the manuscript. Based on this feedback, we have added several new analyses to the revised manuscript (comprising additional 6 supplemental figures, 5 supplemental tables, and 3 supplemental data files). Importantly, we provide new analysis that shows that the model is robust to training and prediction on independent sets of normal samples (Reviewer 1, page 3), new analysis demonstrating the sensitivity limits and specificity (Reviewer 2, page 14), and down-sampling analysis to determine the minimal sequencing requirements of the capture-based NDR assay (Reviewer 2, page 21). In addition, as advised by Reviewer 2, we have moved the “CRC+BRCA” model section to the end of the Results (Fig. 4 and 5 swapped) to improve the clarity of the manuscript. See below for a point-by-point response to the reviewers’ comments and corresponding changes to the manuscript.

Reviewer 1 comments	2
Reviewer 1, General Comments	2
Reviewer 1, Major Comment 1: Spike-Ins / General	2
Reviewer 1, Major Comment 1: Spike-Ins / Relative coverage	2
Reviewer 1, Major Comment 1: Spike-Ins / Independent normals	3
Reviewer 1, Major Comment 1: Spike-Ins / Model coefficients	5
Reviewer 1, Major Comment 2: Data deposition	6
Reviewer 1, Major Comment 3: NDR cfDNA degradation	6
Reviewer 1, Minor Comment 1: typo of in silico	8
Reviewer 1, Minor Comment 2: deep cfDNA WGS sample selection	8
Reviewer 1, Minor Comment 3: confusing claim	9
Reviewer 1, Minor Comment 4: the unit of genomic distances	10
Reviewer 1, Minor Comment 5: the use of colors in the figure	10
Reviewer 2 comments	11
Reviewer 2, General Comments	11
Reviewer 2, Major Comment 1: BRCA tumor-specific NDRs	11
Reviewer 2, Major Comment 2: “CRC+BRCA” model	13
Reviewer 2, Major Comment 3: limit of detection	14
Reviewer 2, Major Comment 4: CNA-based ctDNA fractions	16
Reviewer 2, Major Comment 5: validation of the capture-based assay	17
Reviewer 2, Major Comment 6: ctDNA levels based on diverse approaches	18
Reviewer 2, Major Comment 7: details of the method	20
Reviewer 2, Major Comment 8: coverage recommendation	21
Reviewer 2, Minor Comment 1: targeted sequencing data	23
Reviewer 2, Minor Comment 2: misleading text	24
Reviewer 2, Minor Comment 3: correlations between CRC and CRC+BRCA models	25

Point-by-point response to reviewers

Reviewer 1 comments

Reviewer 1, General Comments

Reviewer Comment	This is a fascinating study that shows that ctDNA burden can be directly predicted (and used as a surrogate for overall tumour burden) using a highly targeted sequencing panel. The study is generally extremely well-conducted, and the work will have very substantial impact on the field. There are three technical issues that should be addressed to strengthen or better support the conclusions that are drawn, but overall I would congratulate the authors on a really nice study!
Author Response	We thank the reviewer for the supportive comments and highly valuable feedback to improve the manuscript.

Reviewer 1, Major Comment 1: Spike-Ins / General

Reviewer Comment	1. Spike-Ins * spiking in a different individual (in silico dilution) has major challenges (SNPs, library prep, etc.)
Author Response	We thank the reviewer for highlighting this aspect. We agree with the reviewer that it is important to consider the potential bias from spike-in/synthetic data. Importantly, we are not using the spike-ins to analyse mutations, so SNPs are likely less of a concern in our analysis. Next, all cfDNA WGS samples (healthy and cancer) were prepared in the same lab using the same library preparation and analysis pipelines. Furthermore, we have validated the model in an independent cohort of 53 colorectal cancer patients, using a capture-based NDR assay, speaking towards the robustness of the approach. To further address the reviewer's questions, we below include detailed answers for the specific highlighted questions: 1) how was relative coverage defined, 2) use of independent normals for training and validation, and 3) make the final coefficients available.

Reviewer 1, Major Comment 1: Spike-Ins / Relative coverage

Reviewer Comment	1. Spike-Ins * unclear exactly how "relative coverage" was defined
------------------	---

Author Response	We thank the reviewer for highlighting this. The new figure below demonstrates how “relative coverage” was defined. For a given promoter region (-150 to 50bp relative to TSS), the raw coverage was divided by the mean raw coverage of the upstream (-2000 to -1000bp relative to TSS) and downstream (1000 to 2000bp relative to TSS) flanks. Thus, the mean coverage of the up and downstream 2 kbp flanks serves as a “normalization factor”. A similar approach was used for exon-intron junctions (see Figure).  Fig. S13 Genomic regions over promoters (top) and first exon-intron junction (bottom) used to calculate relative coverage. The mean coverage of the up and downstream 2kbp flanks (blue) is used as a “normalization factor” for the region of interest (red).
Changes to manuscript	We have added the above Fig. S13 to the supplementary material. Additionally, we have revised and improved the description of the Relative Coverage metric in the Methods: [Page 25 line 20, excerpt] “Read coverage at promoter and junction regions was computed from BAM files with SAMtools depth function. For the promoter region (-150 to 50bp relative to TSS), the mean raw coverage across the region was divided (yielding “relative coverage”) by the mean coverage of the upstream (-2000~-1000 bp relative to TSS) and downstream (1000~2000 bp relative to TSS) flanks (Fig. S13). Thus, the mean coverage of the combined upstream and downstream 2k bp flanks serves as a “normalization factor”. A similar approach was used for exon-intron junctions (Fig. S13).”

Reviewer 1, Major Comment 1: Spike-Ins / Independent normals

Reviewer Comment	1. Spike-Ins * it appears that the same normals were used in both the training & testing datasets, which leads to contamination of training & testing datasets * the validation set is going to require new in silico spike-ins with normal samples not used in the original study
--

Author
Response

The reviewer raises an important point about potential bias from not having independent normal samples in the test set. To evaluate this further, we split the individual normal samples evenly into 2 sets. The first set (N1) was used to perform in silico spike-ins/dilution of the training set, and the second set (N2) was used for in silico dilution of the test set. Briefly, we re-trained the coefficients of the CRC model using the 6 features in Table 1 using the training data (diluted with N1), and we then evaluated the model accuracy on the withheld test samples (diluted with N2). We repeated this procedure 10 times and evaluated the model accuracy on the test set generated using the independent normal samples. Reassuringly, this analysis showed robust model performance across these different training/test splits of healthy individuals (Test set: median Pearson $r = 0.92$; median Spearman correlation, $\rho = 0.93$; median MAE = 5.3%), as shown below in Fig. S5.

Fig. S5 Model performance on 10 test sets generated using independent healthy samples. Individual healthy samples were evenly split into 2 sets, used to dilute training and test sets separately. (a) The correlation (Pearson and Spearman) between the predicted and observed ctDNA fractions across the 10 test sets. (b) The mean absolute error (MAE) between the predicted and observed ctDNA fractions for the 10 test sets.

Additionally, we would like to emphasize that the trained model in the manuscript (Fig. 3) accurately predicted the ctDNA levels in 53 unseen/independent CRC patient plasma samples (Fig. 4b, and c, shown below), speaking towards the model robustness when applied to independent patient samples.

Fig. 4 Targeted NDR assay to quantify ctDNA burden and monitor cancer progression. (a) Schematic showing how targeted NDR sequencing, low-pass WGS, and targeted gene sequencing was performed on a cohort of 53 CRC plasma samples. (b) Comparison of ctDNA fractions inferred by targeted NDR-sequencing and low-pass WGS (ichorCNA). (c) Comparison of ctDNA fractions inferred by

	targeted NDR-sequencing and maximum observed VAFs (maximum VAF of all SNVs identified in a given plasma sample).
Changes to manuscript	We have added the figure above as Fig. S5 in the supplementary to demonstrate the robustness of the model in the presence of spike-ins using independent healthy samples. We have also added paragraphs in the results and methods to describe this analysis. [Page 10 line 10, excerpt] “To further evaluate the robustness of the model when tested on in silico samples generated using independent healthy samples, we generated in silico training and test data using different sets of healthy samples. Reassuringly, this analysis showed robust model performance in the presence of independent train/test healthy samples (Fig. S5; test set median Pearson $r = 0.92$; median Spearman correlation, $\rho = 0.93$; median MAE = 5.3%).” [Methods, Page 28 line 20, excerpt] “To evaluate the robustness of the model when we trained and tested on in silico samples generated using independent healthy samples, we split the normal samples evenly into 2 sets. The first set (N1) was used to perform in silico spike-ins/dilution of the training set, and the second set (N2) was used for in silico dilution of the test set. Briefly, we re-fitted the coefficients of the CRC model (comprising the 6 features in Table 1) using the training data (diluted with the N1 healthy samples), and we then evaluated the model accuracy on the withheld test samples (diluted with N2). We repeated this procedure 10 times and evaluated the model accuracy on the test data generated using the independent normal samples.”

Reviewer 1, Major Comment 1: Spike-Ins / Model coefficients

Reviewer Comment	1. Spike-Ins * please make the final coefficients for the predictive models available in Tables 1 and S5 (or elsewhere as appropriate -- every model needs its coefficients reported for reproducibility)
Author Response	We thank the reviewer for this suggestion. We have added the final coefficients of the CRC and CRC+BRCA models to the new Table S4 in the supplementary material.
Changes to manuscript	A new Table S4 has been added to the Supplementary.

CRC model							Columns
Feature	Gene	Transcript	Region	Group	Coefficient		
1	SHKBP1	ENST00000569716	junction	blood	0.607		Gene : gene name
2	ACSL1	ENST00000454703	junction	blood	0.431		Transcript : transcript ID
3	BCAR1	ENST00000162330	junction	tumor	-0.321		Region : location of nucleosome-depleted site
4	RAB25	ENST00000361084	promoter	tumor	-0.213		Group : gene group based on its expression in blood and tumor
5	PRTN3	ENST00000234347	promoter	blood	0.062		Coefficient: value of the regression coefficient. The intercept value is 0.4368.
6	LSR	ENST00000605618	promoter	tumor	-0.174		

CRC+BRCA model							Columns
Feature	Gene	Transcript	Region	Group	Coefficient		
1	SLC11A1	ENST00000465984	promoter	blood	0.150		Gene : gene name
2	NLRP12	ENST00000324134	promoter	blood	0.181		Transcript : transcript ID
3	PRTN3	ENST00000234347	promoter	blood	0.124		Region : location of nucleosome-depleted site
4	HMB3	ENST00000392841	promoter	blood	0.251		Group : gene group based on its expression in blood and tumor
5	LILRB3	ENST00000460208	promoter	blood	0.140		Coefficient: value of the regression coefficient. The intercept value is -1.3719.
6	ACSL1	ENST00000513001	junction	blood	0.106		
7	GP9	ENST00000307395	junction	blood	0.251		
8	MX2	ENST00000398632	promoter	blood	0.106		
9	RASGRP4	ENST00000615340	promoter	blood	0.222		
10	ATG16L2	ENST00000542481	promoter	blood	0.166		

Table S4 Coefficients for the selected NDRs in the trained models.

Reviewer 1, Major Comment 2: Data deposition

Reviewer Comment	A significant fraction of the value in this study derives from the data it generates. This must be made available, with appropriate meta-data, in a suitable public repository.
Author Response	We completely agree with the reviewer. Our dataset will be made publicly available upon publication. We have created a new EGA project/study (Accession number EGAS00001004657) to store the NGS data with all relevant metadata.
Changes to manuscript	A section on “ Data Availability ” has been added to the manuscript. [Page 29 line 18, excerpt] “Sequencing data of healthy and cancer cfDNA samples are available from the European genome-phenome archive database under the accession code EGAS00001004657.”

Reviewer 1, Major Comment 3: NDR cfDNA degradation

Reviewer Comment	The authors nicely show that expression is one correlate of this, but do not explore other possibilities. The authors need to do a systematic evaluation of other factors as best as possible. They may find nothing, but it is not appropriate, given the scope of this work, to leave that entirely to future studies.
Author Response	The reviewer raises an interesting question about the different factors affecting cfDNA degradation. The underlying biology of cfDNA degradation remains poorly understood (Heitzer et al., Trends Mol Med, 2020). In the introduction of the manuscript, we mention that previous studies have shown

that the size distribution of cfDNA fragments has a mode of ~166bp, suggesting that nucleosome-bound DNA fragments are preserved during cell death and shed into the circulation (Fan et al., PNAS, 2008; Lo et al., Sci Transl Med, 2010). Nucleosome depleted regions (NDRs) are therefore more frequently degraded, yielding a nucleosome-dependent degradation footprint in cfDNA profiles, which can be used to infer tissue of origin (Snyder et al., Cell, 2016; Sun et al., Genome Res, 2019; Cristiano et al., Nature, 2019) and infer qualitative aspects of tumor gene expression (Snyder et al., Cell, 2016; Ulz et al., Nat Genet, 2016). Building on these insights, the main contribution of our work is the development of a novel approach for ctDNA burden estimation.

To further address the reviewer's comment, we performed a new analysis exploring the association between relative coverage and a range of epigenetic features (including gene expression). This analysis showed in addition to gene expression, features negatively correlated with relative coverage of NDRs include DNase hypersensitivity sites, H3K4me3, and H3K27ac, which are markers of open chromatin, active promoters, and active enhancers respectively (Lawrence et al., Trends Genet, 2016). Epigenetic features positively correlated with relative coverage include H3K36me3 and H3K9me3, which are markers of gene bodies and heterochromatin respectively.

Fig. S2 Correlation between relative coverage of NDRs and epigenetic features. For each candidate covariate/predictor, a linear regression is fitted with relative coverage as the response. Whole blood gene expression (fpkm) is binned into 6 bins [unexpressed, $0.01 < \text{fpkm} \leq 0.1$, $0.1 < \text{fpkm} \leq 1$, $1 < \text{fpkm} \leq 5$, $5 < \text{fpkm} \leq 30$, $\text{fpkm} > 30$] and fitted as a categorical covariate with the unexpressed group as the reference group. Peak files of epigenetic features [DNase, H3K4me3, H3K36me3, H3K27ac, H3K4me1, H3K9me3 and H3K27me3] from primary T-cells (E034) were obtained from the Roadmap Epigenomics Project. Epigenetic features are fitted as binary covariates with no signal as the reference group. Barplots of the correlation (r , square root of R^2 multiplied by the coefficient sign) between each feature and relative coverage for (a) Promoter NDRs and (b) Junction NDRs are shown.

Changes to manuscript	We have added the above Fig. S2 and described this new analysis in the results: [Page 6 line 15, excerpt] “To further explore the factors affecting cfDNA degradation at NDRs, we explored the association between NDR relative coverage and a range of epigenetic features (Fig. S2). In addition to gene expression levels, relative coverage was negatively correlated with DNase hypersensitivity, H3K4me3, and H3K27ac, which are markers of open chromatin, active promoters, and active enhancers respectively (Lawrence et al., Trends Genet, 2016). In contrast, H3K36me3 and H3K9me3, markers of gene bodies and heterochromatin, were positively correlated with NDR relative coverage.”
--

Reviewer 1, Minor Comment 1: typo of in silico

Reviewer Comment	in silico should not be hyphenated
Author Response	We thank the reviewer for pointing out this typo.
Changes to manuscript	We have corrected all instances of this error in the paper, including Fig. 3 and 5 .

Reviewer 1, Minor Comment 2: deep cfDNA WGS sample selection

Reviewer Comment	please be clear about exactly how all deep cfDNA WGS samples were selected?
Author Response	We apologize for the lack of details on this important step. Our selection strategy for the CRC samples was briefly mentioned in the results section: “we first used a targeted sequencing panel to screen plasma samples from CRC patients for cases of high ctDNA burden (VAF > 15% for known cancer driver mutations, Fig. 1)”. Similarly, we selected 10 BRCA plasma samples of high ctDNA burden, with either VAF > 15% based on a panel of 77 genes (Table S13) of known mutations (Supplementary Data 2) or significant proportions (>20%) of short cfDNA fragments below 150 bp (Table S1). It has been reported that short fragments below 150 bp are more enriched in high-

	ctDNA plasma samples than normal samples (Mouliere et al., Sci Transl Med, 2018).
Changes to manuscript	We have added the VAF values used for sample selection in Table S1 in the supplementary material. Furthermore, we have improved the description of the sample selection in the manuscript as shown below. [Page 4 line 15, excerpt] “We performed targeted gene sequencing on colorectal and breast cancer plasma samples and selected samples with high SNV VAFs (indicating high ctDNA burden) for deep ~90x cfDNA WGS (Table S1).” [Methods: Page 23 line 14, excerpt] ”We first used a targeted sequencing panel (Table S7) to screen plasma samples from CRC patients and selected 12 samples (Table S1) of likely high ctDNA burden, having maximum VAF > 15% for known CRC cancer driver mutations (Supplementary Data 1). Similarly, we selected 10 BRCA plasma samples of high ctDNA burden, with either VAF > 15% based on a panel of 77 genes (Table S13) of common breast cancer mutations (Supplementary Data 2), or alternatively, significant proportions (>20%) of short (length < 150bp) cfDNA fragments (Table S1). It has been reported that short cfDNA fragments below 150 bp are enriched in high-ctDNA plasma samples (Mouliere et al., Sci Transl Med, 2018). Deep WGS (~90x) was performed on the 12 cfDNA samples from 7 CRC patients and 10 cfDNA samples from 10 BRCA patients.”

Reviewer 1, Minor Comment 3: confusing claim

Reviewer Comment	Page 3, line 6: the claim is made that only clonal mutations can be detected. No data is given to justify the idea that subclonal mutations are undetectable by ctDNA, and this is highly unlikely to be universally true
Author Response	We thank the reviewer for highlighting this point and we agree that this statement is too strong. The problem with subclonal mutations (present only a subset of cancer cells) is that they will likely only be present in a subset of the cancer DNA fragments released into the blood (assuming the uniform probability of shedding across all cancer cells). However, one could theoretically imagine a situation where only a specific tumor subclone releases cancer DNA fragments. In such a situation, mutations in this tumor subclone would be taken as clonal in the blood, and they could indeed be

	used to estimate the ctDNA burden. To avoid misunderstandings, we have relaxed this statement in the manuscript.
Changes to manuscript	We have revised this section in the paper as shown below. [Page 3 line 7, excerpt] “ctDNA burden estimation based on SNVs may therefore be challenging when no clonal mutations exist among the targeted genes.”

Reviewer 1, Minor Comment 4: the unit of genomic distances

Reviewer Comment	please use bp (not b) for genomic distances throughout
Author Response	We thank the reviewer for this comment.
Changes to manuscript	We have corrected “b” to “bp” throughout the manuscript.

Reviewer 1, Minor Comment 5: the use of colors in the figure

Reviewer Comment	please do not use red & green together in the same plot
Author Response	We thank the reviewer for the suggestion.
Changes to manuscript	We have changed the colors accordingly in Fig. 2, 4, and S1 .

Reviewer 2 comments

Reviewer 2, General Comments

Reviewer Comment	The manuscript by Zhu et al. describes a novel method for quantifying plasma ctDNA burden in advanced cancer patients using a small set of tumour/blood-specific nucleosome depleted regions (NDRs). The study takes advantage of the rationale reported previously (Snyder et al. Cell 2016; Ulz et al. Nat Genet 2016), demonstrating the relationship between cfDNA coverage at NDRs and the expression levels in the tumour/blood cells. In addition to the assessment of cfDNA coverage in promoter regions, a further development represents the exploration of sequence coverage across gene bodies, specifically at exon-intron junctions, and its translation into designing a small sequencing panel to monitor disease progression in CRC patients. The idea of designing a capture-based sequencing assay targeting tissue-specific NDRs is novel and represents a valuable, and potentially clinically relevant development. The authors first build a “CRC model” for estimating ctDNA burden from the relative coverage of selected tumour/blood-specific NDRs. They then exemplify its diagnostic value in a cohort of 53 advanced CRC patients (with average VAF = ~30%), in addition to applying the panel to longitudinally collected plasma samples of 5 CRC patients. However, to fully demonstrate its analytical performance and clinical utility, some aspects of the study should be clarified. The manuscript might benefit from implementing suggested comments
Author Response	We thank the reviewer for the supportive feedback and constructive comments. Below we have addressed the specific comments point-by-point.

Reviewer 2, Major Comment 1: BRCA tumor-specific NDRs

Reviewer Comment	First, the study includes two sets of data for CRC and BRCA patients, but in fact only “CRC model” has been fully investigated. The development of “BRCA model” was not reported and, as a result, did not translate into designing a panel assay. It should be clarified whether there were any limitations in identification of “tumour-specific NDRs” in BRCA dataset that could potentially limit the application of NDR-based approach in other cancer types.
Author Response	This is a great question. Our study was originally focused on colorectal cancer (CRC). Here we developed a predictive model, tested it on independent samples (Fig. 3), and finally designed and validated the targeted NDR sequencing approach on an unseen cohort of CRC patient samples (Fig. 4). Subsequently, we wondered if it would be possible to develop a

model that could predict across two distinct tumor types, which led us to develop and analyze the joint CRC+BRCA model (Fig. 5).

To further address the reviewer's comments, we have performed a new analysis to explore the feasibility of developing a BRCA-specific model, without the use of CRC data. We first shortlisted 1894 candidate BRCA tumor-specific transcripts ($\text{fPKM}_{\text{BRCA}} > 10$, $\text{fPKM}_{\text{blood}} < 1$). Using these input features, we then developed a BRCA-specific model, using the same machine learning approach as reported in the manuscript. We split the BRCA cfDNA data into the same training and test sets used to build the "CRC+BRCA" model. As shown in the figure below, a model using the top-8 predictive features (BRCA tumor-specific NDRs) shows high concordance between the predicted and observed ctDNA fractions in the test set (Pearson $r = 0.97$; Spearman correlation, $\rho = 0.97$; MAE=6.1%). This BRCA-specific model has comparable prediction accuracy to the "CRC+BRCA" model using only blood-specific NDRs (Fig. 5, test set BRCA samples: Pearson $r = 0.93$; Spearman correlation, $\rho = 0.96$; MAE=6.1%).

Fig. S10 A BRCA model using BRCA tumor-specific NDRs. (a) The list of top BRCA tumor-specific NDRs that were used for ctDNA content prediction. (b,c) Comparison of observed (in silico simulation) and predicted ctDNA fractions across the BRCA cfDNA samples in the test set.

Changes to manuscript

We have added the above figure as **Fig. S10** in the supplemental material. We have further commented on the performance of a model using BRCA-specific NDRs in the manuscript:

[Page 18 line 12, excerpt]

	“In the unseen CRC and BRCA test data (Fig. 5d), the model achieved an overall accuracy (MAE=4.3%; Pearson $r = 0.95$; Spearman correlation, $\rho = 0.97$; Fig. S9), comparable to the CRC-specific model applied to CRC data (MAE=3.4%; Pearson $r = 0.96$; Spearman correlation, $\rho = 0.97$; Fig. 3e, f) and a BRCA-specific model applied to the BRCA data (MAE=6.1%; Pearson $r = 0.97$; Spearman correlation, $\rho = 0.97$; Fig. S10).”
--	--

Reviewer 2, Major Comment 2: “CRC+BRCA” model

Reviewer Comment	Instead, the authors built a “CRC+BRCA model” limited to only “blood-specific” NDRs, however without more details on different tumour types used for building this model (referred to as 20 different solid tumour types), and also without demonstrating its clinical use by designing a targeted panel assay. It is not clear whether some of the blood-specific NDRs identified in “CRC+BRCA model” overlap with the previous CRC model. This section of the manuscript (“CRC+BRCA model”) brings a confusion in understanding the flow of the document, since in the following section only the CRC model is applied. A suggestion would be to move this part at the end of the Result section. Regardless, developing “a universal NDR approach” across different cancer types using the presented “blood-specific model”, if applied across a variety of cancer types, would have had a much higher impact in the field in comparison with being limited to one cancer type. Justifying the selection of a particular design should be provided.
Author Response	We thank the reviewer for the great suggestion and have moved the “CRC+BRCA” model part to the end of the Results section in the revised manuscript. We completely agree with the reviewer that a universal NDR model across multiple cancer types would have a much higher impact compared with cancer-specific models. We also agree that our manuscript merely outlines the possibility of such a pan-cancer approach and that studies across even larger patient cohorts and other cancer types are needed to test the robustness of such an approach. Accordingly, we have moved the CRC+BRCA model development section to the end of the manuscript, to better emphasize that additional work is needed in this area. We have also further emphasized this point in the revised manuscript. To further address the reviewer’s questions about the overlap of predictive NDRs in the CRC and CRC+BRCA models, we note that 2 genes (ACSL1 and PRTN3, shown in the Table below) overlapped between the 6-feature CRC and 10-feature CRC+BRCA models (see also Table 1 and S11):

	CRC model    Feature Gene Transcript Region Group Coefficient Columns     1 SHKBP1 ENST00000569716 junction blood 0.607 Gene : gene name   2 ACSL1 ENST00000454703 junction blood 0.431 Transcript : transcript ID   3 BCAR1 ENST00000162330 junction tumor -0.321 Region : location of nucleosome-depleted site   4 RAB25 ENST00000361084 promoter tumor -0.213 Group : gene group based on its expression in blood and tumor   5 PRTN3 ENST00000234347 promoter blood 0.062 Coefficient: value of the regression coefficient. The intercept value is 0.4368.   6 LSR ENST00000605618 promoter tumor -0.174     CRC+BRCA model    Feature Gene Transcript Region Group Coefficient Columns     1 SLC11A1 ENST00000465984 promoter blood 0.150 Gene : gene name   2 NLRP12 ENST00000324134 promoter blood 0.181 Transcript : transcript ID   3 PRTN3 ENST00000234347 promoter blood 0.124 Region : location of nucleosome-depleted site   4 HMB3 ENST00000392841 promoter blood 0.251 Group : gene group based on its expression in blood and tumor   5 LILRB3 ENST00000460208 promoter blood 0.140 Coefficient: value of the regression coefficient. The intercept value is -1.3719.   6 ACSL1 ENST00000513001 junction blood 0.106    7 GP9 ENST00000307395 junction blood 0.251    8 MX2 ENST00000398632 promoter blood 0.106    9 RASGRP4 ENST00000615340 promoter blood 0.222    10 ATG16L2 ENST00000542481 promoter blood 0.166     Table S4 Coefficients for the selected NDRs in the trained models.	Feature	Gene	Transcript	Region	Group	Coefficient	Columns	1	SHKBP1	ENST00000569716	junction	blood	0.607	Gene : gene name	2	ACSL1	ENST00000454703	junction	blood	0.431	Transcript : transcript ID	3	BCAR1	ENST00000162330	junction	tumor	-0.321	Region : location of nucleosome-depleted site	4	RAB25	ENST00000361084	promoter	tumor	-0.213	Group : gene group based on its expression in blood and tumor	5	PRTN3	ENST00000234347	promoter	blood	0.062	Coefficient: value of the regression coefficient. The intercept value is 0.4368.	6	LSR	ENST00000605618	promoter	tumor	-0.174		Feature	Gene	Transcript	Region	Group	Coefficient	Columns	1	SLC11A1	ENST00000465984	promoter	blood	0.150	Gene : gene name	2	NLRP12	ENST00000324134	promoter	blood	0.181	Transcript : transcript ID	3	PRTN3	ENST00000234347	promoter	blood	0.124	Region : location of nucleosome-depleted site	4	HMB3	ENST00000392841	promoter	blood	0.251	Group : gene group based on its expression in blood and tumor	5	LILRB3	ENST00000460208	promoter	blood	0.140	Coefficient: value of the regression coefficient. The intercept value is -1.3719.	6	ACSL1	ENST00000513001	junction	blood	0.106		7	GP9	ENST00000307395	junction	blood	0.251		8	MX2	ENST00000398632	promoter	blood	0.106		9	RASGRP4	ENST00000615340	promoter	blood	0.222		10	ATG16L2	ENST00000542481	promoter	blood	0.166	
Feature	Gene	Transcript	Region	Group	Coefficient	Columns																																																																																																																									
1	SHKBP1	ENST00000569716	junction	blood	0.607	Gene : gene name																																																																																																																									
2	ACSL1	ENST00000454703	junction	blood	0.431	Transcript : transcript ID																																																																																																																									
3	BCAR1	ENST00000162330	junction	tumor	-0.321	Region : location of nucleosome-depleted site																																																																																																																									
4	RAB25	ENST00000361084	promoter	tumor	-0.213	Group : gene group based on its expression in blood and tumor																																																																																																																									
5	PRTN3	ENST00000234347	promoter	blood	0.062	Coefficient: value of the regression coefficient. The intercept value is 0.4368.																																																																																																																									
6	LSR	ENST00000605618	promoter	tumor	-0.174																																																																																																																										
Feature	Gene	Transcript	Region	Group	Coefficient	Columns																																																																																																																									
1	SLC11A1	ENST00000465984	promoter	blood	0.150	Gene : gene name																																																																																																																									
2	NLRP12	ENST00000324134	promoter	blood	0.181	Transcript : transcript ID																																																																																																																									
3	PRTN3	ENST00000234347	promoter	blood	0.124	Region : location of nucleosome-depleted site																																																																																																																									
4	HMB3	ENST00000392841	promoter	blood	0.251	Group : gene group based on its expression in blood and tumor																																																																																																																									
5	LILRB3	ENST00000460208	promoter	blood	0.140	Coefficient: value of the regression coefficient. The intercept value is -1.3719.																																																																																																																									
6	ACSL1	ENST00000513001	junction	blood	0.106																																																																																																																										
7	GP9	ENST00000307395	junction	blood	0.251																																																																																																																										
8	MX2	ENST00000398632	promoter	blood	0.106																																																																																																																										
9	RASGRP4	ENST00000615340	promoter	blood	0.222																																																																																																																										
10	ATG16L2	ENST00000542481	promoter	blood	0.166																																																																																																																										
Changes to manuscript	Following the reviewer’s advice, we have moved the “CRC+BRCA” model part to the end of the Result section in the revised manuscript. We have also emphasized the lack of validation for the pan-cancer model in the revised manuscript. [Discussion Page 22 line 1, excerpt] “... tumor types and metastatic lesions. However, studies across even larger patient cohorts and multiple cancer types are needed to further test and establish the robustness of such an approach.”																																																																																																																														

Reviewer 2, Major Comment 3: limit of detection

Reviewer Comment	Second, the authors report on improved detection in “low ctDNA burden” cases using NDR-based approach, however, without addressing the levels of sensitivity/limit of detection. First, the terms “high ctDNA burden” and “low ctDNA burden” should be clarified. ctDNA levels in advanced cancer patients can be high based on SNV analysis, but in case these tumours are not driven by SCNA, ichorCNA would identify them as “low ctDNA burden”. The differences should be clearly defined in the manuscript.
Author Response	We agree with the reviewer that the lower limit of detection (LOD) is an important parameter to determine. To address this question, we conducted a new LOD analysis using the approach adopted by ichorCNA (Adalsteinsson et al., Nat Commun, 2017). Briefly, we used the 113 in silico CRC samples in the test set (Fig. 3e) as true positives and 40 random subsets (~80x each) from the healthy data as true negatives. At a set threshold of $\leq 2\%$ ctDNA fraction, our model correctly predicted all cancer (100% sensitivity) and healthy samples (100% specificity, Table S5). In comparison, at the threshold $\leq 1\%$, the sensitivity was maintained at 100%, but the specificity dropped to 75%. Thus, we recommend 2% as the lower limit of tumor detection for the CRC NDR model.

Similarly, we evaluated the LOD for the CRC+BRCA model by using the 206 in silico samples (113 CRC + 93 BRCA) in the test set (Fig. 5d) as true positives and the 40 healthy subsets above as true negatives. This data highlighted a LOD of 3% in order to achieve 100% sensitivity and 100% specificity in the CRC+BRCA model (Table S12).

Threshold	CRC model		CRC+BRCA model	
	sensitivity	specificity	sensitivity	specificity
1%	100%	75.0%	99.5%	37.5%
2%	100%	100%	99.5%	87.5%
3%	100%	100%	100%	100%

To further address the reviewer’s comment about low/high ctDNA burden, we agree that this terminology can be ambiguous for SNV and SCNA-based estimates. In the manuscript, we use “low/high ctDNA burden” as a “general” term referring to the relative fraction of ctDNA molecules out of all cfDNA molecules in the plasma sample. We have improved the description of this definition in the revised manuscript. Finally, we would like to emphasize that the CRC NDR model is trained using consensus ctDNA burden estimates, where ctDNA burden is estimated using both SNV (absCN-Seq, PurBayes) and SCNA-based (Theta2, TitanCNA) methods applied to the deep WGS cfDNA samples.

Changes to manuscript

We have added **Table S5** and **S12**, providing predicted ctDNA fractions for the positive and negative samples in the LOD analysis above for the CRC and CRC+BRCA models, respectively. We have also added new sections in the revised manuscript to describe the new LOD analysis:

[Page 10 line 3, excerpt]

“Next, we explored the lower limit for ctDNA detection in the NDR model. Using a previous approach (Adalsteinsson et al., Nat Commun, 2017), we evaluated the sensitivity and specificity of the model as a function of ctDNA fraction threshold. We used the 113 in silico test set CRC samples (Fig. 3e) as positives, and 40 random subsets (~80x each) from the healthy data as negatives. At a 2% ctDNA fraction threshold, the model correctly predicted all positive and negative samples (100% sensitivity and specificity, Table S5). In comparison, at a 1% threshold, the sensitivity was maintained at 100% but the specificity dropped to 75%.”

[Page 18 line 18, excerpt]

“We analyzed the lower limit of detection for the CRC+BRCA model by evaluating the sensitivity and specificity of the model as a function of ctDNA fraction threshold. We used the 206 in silico test set samples (113 CRC + 93 BRCA, Fig. 5d) as positives and

	40 healthy random subsets (~80x) as negatives. At a 3% ctDNA fraction detection limit, the CRC+BRCA model achieved 100% sensitivity and specificity (Table S12). In comparison, at a 2% threshold, the sensitivity was almost maintained at 99.5% but the specificity dropped to 88%.” Finally, we have added a section in the revised manuscript to clarify our use of the terms low/high ctDNA burden: [Page 4 line 2, excerpt] “ctDNA burden refers to the relative amount of ctDNA out of all cfDNA molecules in a plasma sample.”
--	---

Reviewer 2, Major Comment 4: CNA-based ctDNA fractions

Reviewer Comment	This is in particularly relevant in the section “Quantitative estimation of colorectal cancer ctDNA burden”, where the cases were pre-selected (and in silico diluted) based on the VAFs by SNV analysis, without considering their SCNA profiles. Therefore, the lower detection rates by ichorCNA could have possibly been biased just by the fact that these samples had low extent of SCNA (while still having high ctDNA levels identified by SNV analysis, Fig. S4). Can the authors add ichorCNA-based ctDNA fractions for CRC cases in Table S1 to assess this?
Author Response	We thank the reviewer for this suggestion. We have added ichorCNA-based predicted ctDNA fractions for the deep-WGS CRC samples in Table S1. Firstly, as shown in Table S1, the “observed/true” ctDNA fractions of the CRC deepWGS cfDNA samples are consensus estimates based on four different methods: THetA2 and TitanCNA are CNA-based, and AbsCN-seq and PurBayes are SNV-based. As expected, all 12 CRC samples have high ichorCNA-based ctDNA fractions (median ~37%), comparable with the “observed” consensus estimates (median ~48%). However, ichorCNA detection rates are low for in silico generated samples with ctDNA fractions <5% as shown in Fig. S7. For example, as shown below, patient CRC-11 had similar ctDNA fractions estimated with the consensus approach and ichorCNA (61% and 60% respectively), but the 3 in silico diluted samples (ctDNA = 0.5%, 1%, 1.5%) were wrongly classified as healthy (ctDNA = 0%) by ichorCNA. Importantly, these results are consistent with the reported lower limit of detection of 3% for ichorCNA (Adalsteinsson et al., Nat Commun, 2017).

Fig. S6a Comparison of observed (in silico simulation) and ichorCNA-predicted ctDNA fractions across the CRC ctDNA samples. Three low-ctDNA samples for CRC-11 were predicted as non-cancerous by ichorCNA, highlighted in red.

Changes to manuscript

We have added ichorCNA-based predicted ctDNA fractions for the CRC samples in **Table S1**.

sample ID	cancer type	coverage	tumor content estimation - repetition 1				tumor content estimation - repetition 2				tumor content estimation - repetition 3				mean ctDNA content	median ctDNA content	ichorCNA-estimated ctDNA content	median VAF
			theta2	titanCNA	abscnseq	PurBayes	theta2	titanCNA	abscnseq	PurBayes	theta2	titanCNA	abscnseq	PurBayes				
CRC-1	CRC	80.84	0.38	0.55	0.48	0.43	0.38	0.55	0.48	0.43	0.38	0.55	0.48	0.43	0.45	0.45	0.40	0.29
CRC-9	CRC	84.67	0.43	0.65	0.50	NA	0.43	0.65	0.50	NA	0.43	0.63	0.50	0.33	0.56	0.50	0.41	0.53
CRC-10	CRC	95.22	0.66	0.49	NA	0.40	0.66	0.49	NA	0.40	0.66	0.49	NA	0.40	0.52	0.49	0.33	0.41
CRC-2	CRC	90.65	0.86	NA	NA	0.93	0.86	0.82	NA	0.94	0.86	0.81	NA	0.93	0.87	0.86	0.54	0.79
CRC-3	CRC	100.63	0.67	0.46	0.32	0.30	0.75	0.48	0.32	0.83	0.75	0.50	0.32	0.79	0.54	0.49	0.21	0.29
CRC-11	CRC	79.96	0.64	0.69	0.49	0.44	0.64	0.69	0.49	0.44	0.59	0.68	0.49	0.88	0.60	0.61	0.60	0.61
CRC-12	CRC	79.90	0.27	0.51	0.44	0.35	0.27	0.51	0.44	0.35	0.27	0.52	0.56	0.35	0.40	0.39	0.38	0.42
CRC-4	CRC	87.27	0.30	0.40	0.35	0.36	0.30	0.40	0.35	0.36	0.30	0.43	0.35	0.36	0.35	0.35	0.29	0.19
CRC-5	CRC	75.76	0.45	0.64	0.59	0.44	0.45	0.64	0.59	0.44	0.45	0.63	0.59	0.94	0.57	0.59	0.46	0.54
CRC-6	CRC	72.56	0.35	0.59	NA	NA	0.35	0.59	NA	NA	0.35	0.60	NA	0.47	0.47	0.47	0.36	0.73
CRC-7	CRC	71.63	0.25	0.69	0.50	0.36	0.25	0.69	0.50	0.36	0.25	0.47	0.50	0.37	0.43	0.42	0.23	0.23
CRC-8	CRC	76.72	0.34	0.54	0.51	0.37	0.34	0.54	0.51	0.37	0.34	0.54	0.51	0.37	0.44	0.44	0.17	0.26

sample ID	cancer type	coverage	ichorCNA-estimated ctDNA content	median VAF	fraction of ctDNA fragments <150bp (%)
BRCA-1	BRCA	86.98	0.70	0.11	21.97
BRCA-2	BRCA	79.41	0.76	0.37	29.35
BRCA-3	BRCA	94.33	0.13	0.20	27.23
BRCA-4	BRCA	89.86	0.66	0.25	22.74
BRCA-5	BRCA	87.92	0.59	0.17	29.61
BRCA-6	BRCA	85.45	0.51	0.06	24.68
BRCA-7	BRCA	78.51	0.54	0.32	31.02
BRCA-8	BRCA	76.67	0.74	0.01	26.47
BRCA-9	BRCA	71.80	0.41	0.46	27.02
BRCA-10	BRCA	71.50	0.54	0.18	32.85

Table S1. ctDNA burden estimation of plasma samples from cancer patients.

Reviewer 2, Major Comment 5: validation of the capture-based assay

Reviewer Comment	The authors report on a quantitative nature of their methodology, and thus should demonstrate the performance validation of the capture-based assay, and determine the levels of sensitivity. Although in silico dilutions of a limited number of WGS samples were prepared while building the model, the lower limits of ctDNA detection were not reported for the actual capture-based sequencing assay. The authors highlighted the clinical application of their method in a cohort of CRC patients, however to address the specificity of the capture-based assay, the approach would further require validation in a cohort of healthy individuals.
Author Response	We agree with the reviewer that methods for the detection of cancer should be evaluated on healthy samples. Above (comment 3), we outlined that the lower limit of detection is likely in the 1-2% range for the CRC model, meaning

	that below this ctDNA threshold, it is challenging for the NDR model to distinguish between cancer and healthy samples:    Threshold CRC model CRC+BRCA model   sensitivity specificity sensitivity specificity     1% 100% 75.0% 99.5% 37.5%   2% 100% 100% 99.5% 87.5%   3% 100% 100% 100% 100%    We believe this shows that the NDR approach is not well suited for screening for the presence/absence of cancer in a clinical/diagnostic setting, but that the utility instead lies in the joint and low-cost estimation of ctDNA burden and mutational profiles in response to treatment interventions. We have highlighted this point further in the discussion of the revised manuscript. To further address the reviewer’s question, the capture-based NDR assay was applied to 53 unseen CRC samples having a wide range of ctDNA fractions (~0-~50%, Fig. 4). Here, the capture-based NDR estimates were strongly correlated with SNV VAFs and ichorCNA estimates across the range of estimated ctDNA fractions, indicating that the quantitative estimation works well both in the presence of low and high ctDNA levels. Importantly, the 3 samples that were determined as ctDNA-free by the NDR method were also predicted as ctDNA-free based on SNV analysis and ichorCNA (Table S6). Although these 3 samples cannot substitute for known healthy (cancer-free) samples, this concordance does highlight the specificity of the capture-based NDR approach.	Threshold	CRC model		CRC+BRCA model		sensitivity	specificity	sensitivity	specificity	1%	100%	75.0%	99.5%	37.5%	2%	100%	100%	99.5%	87.5%	3%	100%	100%	100%	100%
Threshold	CRC model		CRC+BRCA model																						
	sensitivity	specificity	sensitivity	specificity																					
1%	100%	75.0%	99.5%	37.5%																					
2%	100%	100%	99.5%	87.5%																					
3%	100%	100%	100%	100%																					
Changes to manuscript	We have highlighted the limitation of the NDR-based method for predicting the presence/absence of cancer in the revised manuscript: [Page 21 line 14, excerpt] “While the estimated lower limit of detection (~2%) of the NDR approach is likely not suited for screening of cancer in healthy/cancer-free individuals, the approach could enable low-cost and simultaneous quantitative estimation of ctDNA burden and mutational profiling in response to treatment interventions.”																								

Reviewer 2, Major Comment 6: ctDNA levels based on diverse approaches

Reviewer Comment	The authors suggest to simultaneously profile actionable cancer mutations and quantitatively estimate ctDNA burden. To strengthen the message of the manuscript, the authors should demonstrate the actual benefit of NDR-based method over the other methods. To address this, it would be informative to
------------------	--

	determine the proportion of samples (among 53 CRC patients) detected by NDR-based approach while being SNV-negative, and conversely, to acknowledge the proportion of NDR-negative samples detected by SNV-based analysis. In particular, it would be important to highlight the samples detected by NDR-based analysis that were ichorCNA-negative (which appeared to have no SNVs detected). This section would benefit from providing a comprehensive supplementary data for all 53 patients showing their ctDNA levels based on ichorCNA, alongside the SNV analysis and the NDR-based analysis.																																																																																																																																																														
Author Response	We thank the reviewer for the suggestion to provide the estimated ctDNA fractions based on the NDR method, ichorCNA, and SNV analysis for the 53 CRC samples. We have included these estimates in the new supplementary Table S6. We believe a key benefit of the NDR approach is the potential to have a single joint capture-based assay for both mutation calling and ctDNA burden estimation, reducing the need for a second lp-WGS assay for auxiliary burden estimation. Indeed, capture-based NDR ctDNA burden estimates were strongly correlated with both ichorCNA (Pearson $r = 0.84$, Fig. 5) and SNV VAFs ($r = 0.85$). This is especially important in patients without mutations in the captured regions as well as for copy-number stable tumors. Furthermore, it is non-trivial to directly map a VAF to a ctDNA burden estimate, since the VAF has to be corrected for the clonality and zygosity of the variant allele, which can be challenging to estimate with a targeted ctDNA-assay. We have summarized these points in the revised manuscript. To further address the reviewer's question, 4 samples were NDR-negative (N=3) or below the 2% limit of detection (N=1); all 4 samples were also SNV and ichorCNA-negative. Furthermore, out of the remaining 49 NDR-positive samples (ctDNA fraction > 2%), 22 were SNV-negative and 14 were both SNV and ichorCNA-negative. Notably, 6/14 samples had SNVs detected at other timepoints, and in 5 of these 6 samples (83%) we were able to rescue the sample as SNV-positive through inspection of the raw NGS pileup data (Table S15). For example, as described in the manuscript, samples 357_230913 and 1531_111016 were initially called SNV and ichorCNA-negative but NDR-positive. However, we re-analyzed these samples for SNVs that were identified at earlier time-points in the given patients. Indeed, the raw sequencing data supported the presence of the expected SNVs in the samples (Table S9, 'VAF-manual' column).    Patient Sample Day chr pos (GRCh38) Ref Alt Symbol/Gene VAF-Mutect VAF-VarScan VAF-manual ctDNA detection - ichorCNA ctDNA detection - NDR     357 T357_230913 34 chr13 32362670 G T BRCA2 undetected undetected 0 negative positive   chr2 211657762 T G ERBB4 undetected undetected 0   chr12 25245350 C T KRAS undetected undetected 0.0105   chr17 7674945 G A TP53 undetected undetected 0   chr2 43224819-36 - 3CCGCCGC ZFP36L2 undetected undetected 0   1531 T1531_111016 160 chr5 112839439 C A APC undetected undetected 0 negative positive   chr17 7675088 C T TP53 undetected undetected 0.0012   1531 T1531_111116 191 chr5 112839439 C A APC undetected undetected 0 negative negative   chr17 7675088 C T TP53 undetected undetected 0   575 T575_270616 864 chr3 142555897-8 - -T ATR undetected undetected 0.0579 positive positive   chr17 7673811 A G TP53 undetected undetected 0.0546   519 T519_140116 726 chr5 112839465 C T APC undetected undetected 0.0247 positive positive   chr20 58903555 C A GNAS undetected undetected 0.0232   chr5 56871965 G T MAP3K1 undetected undetected 0.0177   chr17 7673803 G A TP53 undetected undetected 0.0448   	Patient	Sample	Day	chr	pos (GRCh38)	Ref	Alt	Symbol/Gene	VAF-Mutect	VAF-VarScan	VAF-manual	ctDNA detection - ichorCNA	ctDNA detection - NDR	357	T357_230913	34	chr13	32362670	G	T	BRCA2	undetected	undetected	0	negative	positive	chr2	211657762	T	G	ERBB4	undetected	undetected	0	chr12	25245350	C	T	KRAS	undetected	undetected	0.0105	chr17	7674945	G	A	TP53	undetected	undetected	0	chr2	43224819-36	-	3CCGCCGC	ZFP36L2	undetected	undetected	0	1531	T1531_111016	160	chr5	112839439	C	A	APC	undetected	undetected	0	negative	positive	chr17	7675088	C	T	TP53	undetected	undetected	0.0012	1531	T1531_111116	191	chr5	112839439	C	A	APC	undetected	undetected	0	negative	negative	chr17	7675088	C	T	TP53	undetected	undetected	0	575	T575_270616	864	chr3	142555897-8	-	-T	ATR	undetected	undetected	0.0579	positive	positive	chr17	7673811	A	G	TP53	undetected	undetected	0.0546	519	T519_140116	726	chr5	112839465	C	T	APC	undetected	undetected	0.0247	positive	positive	chr20	58903555	C	A	GNAS	undetected	undetected	0.0232	chr5	56871965	G	T	MAP3K1	undetected	undetected	0.0177	chr17	7673803	G	A	TP53	undetected	undetected	0.0448
Patient	Sample	Day	chr	pos (GRCh38)	Ref	Alt	Symbol/Gene	VAF-Mutect	VAF-VarScan	VAF-manual	ctDNA detection - ichorCNA	ctDNA detection - NDR																																																																																																																																																			
357	T357_230913	34	chr13	32362670	G	T	BRCA2	undetected	undetected	0	negative	positive																																																																																																																																																			
			chr2	211657762	T	G	ERBB4	undetected	undetected	0																																																																																																																																																					
			chr12	25245350	C	T	KRAS	undetected	undetected	0.0105																																																																																																																																																					
			chr17	7674945	G	A	TP53	undetected	undetected	0																																																																																																																																																					
			chr2	43224819-36	-	3CCGCCGC	ZFP36L2	undetected	undetected	0																																																																																																																																																					
1531	T1531_111016	160	chr5	112839439	C	A	APC	undetected	undetected	0	negative	positive																																																																																																																																																			
			chr17	7675088	C	T	TP53	undetected	undetected	0.0012																																																																																																																																																					
1531	T1531_111116	191	chr5	112839439	C	A	APC	undetected	undetected	0	negative	negative																																																																																																																																																			
			chr17	7675088	C	T	TP53	undetected	undetected	0																																																																																																																																																					
575	T575_270616	864	chr3	142555897-8	-	-T	ATR	undetected	undetected	0.0579	positive	positive																																																																																																																																																			
			chr17	7673811	A	G	TP53	undetected	undetected	0.0546																																																																																																																																																					
519	T519_140116	726	chr5	112839465	C	T	APC	undetected	undetected	0.0247	positive	positive																																																																																																																																																			
			chr20	58903555	C	A	GNAS	undetected	undetected	0.0232																																																																																																																																																					
			chr5	56871965	G	T	MAP3K1	undetected	undetected	0.0177																																																																																																																																																					
			chr17	7673803	G	A	TP53	undetected	undetected	0.0448																																																																																																																																																					

	Table S9 Mutations missed by the SNV callers for the CRC patients with serial plasma samples.
Changes to manuscript	We have provided the estimated ctDNA levels based on the NDR method, ichorCNA, and SNV analysis for the 53 CRC patients in the new supplementary Table S6. Besides, we have added Table S15, providing the information of the samples with NDR-positive ctDNA detection but SNV and ichorCNA-negative. We have further elaborated on the key benefits of a combined mutation and NDR capture-based assay in the discussion of the revised manuscript. [Page 20 line 17, excerpt] “Additionally, absolute ctDNA fraction estimation from SNVs requires co-estimation of allele zygosity and clonality (Carter et al., Nat Biotechnol, 2012), which may be challenging to infer for metastatic patients with multiple independently evolving tumors contributing ctDNA to the blood circulation..... Importantly, an integrated NDR/gene assay would be able to estimate ctDNA burden in patients without clonal mutations in targeted cancer genes, potentially corresponding to 5-70% of patients depending on cancer type (Bailey et al., Cell, 2018). While the estimated lower limit of detection (~2%) of the NDR approach is likely not suited for screening of cancer in healthy/cancer-free individuals, the approach could enable low-cost and simultaneous quantitative estimation of ctDNA burden and mutational profiling in response to treatment interventions.”

Reviewer 2, Major Comment 7: details of the method

Reviewer Comment	The Methods section provides a lack of information on explaining the prediction of ctDNA fractions using NDR-based method. Please add more details. Similarly, a clarification is needed in relation to estimation of the “relative cfDNA coverage score”. Was the relative coverage calculated within -1K to +1K region from TSS or specifically in the regions -150 to 50bp relative to TSS, or -300 to -100bp relative to first exon end?
Author Response	We apologize for the lack of clarity. The new figure below demonstrates how “relative coverage” was defined. For a given promoter region (-150 to 50bp relative to TSS), the raw coverage was divided by the mean raw coverage of the upstream (-2000 to -1000bp relative to TSS) and downstream (1000 to 2000bp relative to TSS) flanks. Thus, the mean coverage of the up and

	downstream 2 kbp flanks serves as a “normalization factor”. A similar approach was used for exon-intron junctions (see Figure).  Fig. S13 Genomic regions over promoters (top) and first exon-intron junction (bottom) used to calculate relative coverage. The mean coverage of the up and downstream 2kbp flanks (blue) is used as a “normalization factor” for the region of interest (red). The collection of these relative coverage values of NDRs across promoters and exon-intron junctions were modelled as features (independent variables) for prediction of ctDNA fractions (dependent variable) using a Lasso linear regression approach (Fig. 3c, Methods paragraph ‘Lasso regularized regression to predict ctDNA fraction’).
Changes to manuscript	We have added the above Fig. S13 to the supplementary material. Additionally, we have revised and improved the description of the Relative Coverage metric for prediction of ctDNA fractions in the Methods: [Page 25 line 210, excerpt] “Read coverage at promoter and junction regions was computed from BAM files with SAMtools depth function. For the promoter region (-150 to 50bp relative to TSS), the mean raw coverage across the region was divided (yielding “relative coverage”) by the mean coverage of the upstream (-2000~-1000 bp relative to TSS) and downstream (1000~2000 bp relative to TSS) flanks (Fig. S13). Thus, the mean coverage of the combined upstream and downstream 2k bp flanks serves as a “normalization factor”. A similar approach was used for exon-intron junctions (Fig. S13).”

Reviewer 2, Major Comment 8: coverage recommendation

Reviewer Comment	The authors highlight that their method is cost-effective but this statement lacks a comparison with other methods. Additionally, could the authors perform in silico experiment to assess the effect of reducing the amount of sequencing on diagnostic performance of NDR-based analysis? At the moment, a coverage 300x is required but could this be further reduced to keep a similar performance?
------------------	---

Author Response	We agree with the reviewer and have replaced the term ‘cost-effective’ with ‘low-cost’ in the revised manuscript. The NDR-based method requires profiling of ≤ 10 NDRs and < 25 kbp of genomic sequence. Targeted cfDNA assays cover hotspots and genes ranging in size upwards of 300-500 genes and > 1000 kbp (FoundationOne Liquid CDx and GuardantOMNI), with larger panels required for relevance across multiple cancer types and tumor burden estimation. Adding targeted NDR profiling to such assays would increase the utility (single assay for mutational profiling and independent ctDNA burden estimation across patients) while only yielding a minor increment to the panel size and total sequencing cost ($< 5\%$, 25kbp/1000kbp). To further address the reviewer’s comment on coverage recommendations, we performed a down-sampling analysis to assess the effects of reduced coverage in the capture-based NDR assay. For each CRC ctDNA sample (N=53, 181~514x, mean=~ 300x), we downsampled reads to construct cfDNA samples of 100x, 50x, 25x, and 10x coverage. Expectedly, concordance with the 300x reference samples decreased with the mean downsampled read coverage. However, cfDNA samples of 100x and 50x were still highly concordant with the original 300x samples, with both Pearson and Spearman correlations > 0.9 (Fig. S12 below). Intriguingly, the sequencing demand of 25 kbp capture-based sequencing at 100x is equivalent to WGS at ~ 0.001x depth ($0.0008 \approx 25 \times 3 / 3 \times 10^9 \times 100$), orders of magnitude lower than the requirements of current lp-WGS approaches such as ichorCNA (~ 0.1x), highlighting the flexibility and low cost of the targeted NDR approach.  Fig. S12 Comparison of the predicted ctDNA fractions in the 53 original cfDNA samples with capture-based NDR sequencing (mean coverage ~ 300x) and their downsampled counterparts (100x, 50x, 25x, and 10x, respectively).
Changes to manuscript	We have added the above Fig. S12 to the supplementary material. We have clarified the cost properties of the NDR approach in the discussion of the revised manuscript: [Page 21 line 5, excerpt] “Since the ctDNA burden estimation model requires data from 10 or less NDRs, these regions can be profiled at low cost by capturing < 25 kbp of genomic sequence. Targeted cfDNA assays often cover hundreds of genes and > 1 Mb captured genomic sequence, with larger panels required for profiling across cancer types and tumour mutation burden estimation (Buchhalter et al.,

	Int J Cancer, 2019). It would be straightforward to co-profile NDRs in such assays, with only a minor increment in panel size. Furthermore, down-sampling analysis showed that the NDR approach is robust down to 100x sequence coverage (Fig. S12), imposing a sequencing demand equivalent to ~0.001x WGS, orders of magnitude lower than current lp-WGS approaches (Adalsteinsson et al., Nat Commun, 2017).”
--	---

Reviewer 2, Minor Comment 1: targeted sequencing data

Reviewer Comment	The authors mentioned that they performed targeted sequencing in breast cancer patients (page 13, line 5), but the data does not seem to be provided. Please clarify this. Additionally, please indicate specific locations corresponding to each gene in the panel (Table S7)																																																																		
Author Response	We apologize that this aspect was not clearly documented. To select BRCA samples (N=10) for deep cfDNA WGS, we initially selected BRCA plasma samples of likely high ctDNA burden, having maximum VAF > 15% based on a panel of 77 genes (Table S13) commonly mutated in breast cancer (Supplementary Data 2). We selected three additional BRCA patient samples with a significant proportion (>20%) of short cfDNA fragments (<150 bp) (Table S1), since it has been reported that short fragments are enriched in high-ctDNA plasma as compared to normal cfDNA samples (Mouliere et al., Sci Transl Med, 2018).    sample ID cancer type coverage ichorCNA-estimated ctDNA content median VAF fraction of cfDNA fragments <150bp (%)     BRCA-1 BRCA 86.98 0.70 0.11 21.97   BRCA-2 BRCA 79.41 0.76 0.37 29.35   BRCA-3 BRCA 94.33 0.13 0.20 27.23   BRCA-4 BRCA 89.86 0.66 0.25 22.74   BRCA-5 BRCA 87.92 0.59 0.17 29.61   BRCA-6 BRCA 85.45 0.51 0.06 24.68   BRCA-7 BRCA 78.51 0.54 0.32 31.02   BRCA-8 BRCA 76.67 0.74 0.01 26.47   BRCA-9 BRCA 71.80 0.41 0.46 27.02   BRCA-10 BRCA 71.50 0.54 0.18 32.85    Table S1 (part) ctDNA burden estimation of plasma samples from cancer patients. In addition, we have added the BED files as Supplementary data 1 and 2, showing the locations corresponding to the gene panels for SNV analysis in CRC and BRCA, respectively.	sample ID	cancer type	coverage	ichorCNA-estimated ctDNA content	median VAF	fraction of cfDNA fragments <150bp (%)	BRCA-1	BRCA	86.98	0.70	0.11	21.97	BRCA-2	BRCA	79.41	0.76	0.37	29.35	BRCA-3	BRCA	94.33	0.13	0.20	27.23	BRCA-4	BRCA	89.86	0.66	0.25	22.74	BRCA-5	BRCA	87.92	0.59	0.17	29.61	BRCA-6	BRCA	85.45	0.51	0.06	24.68	BRCA-7	BRCA	78.51	0.54	0.32	31.02	BRCA-8	BRCA	76.67	0.74	0.01	26.47	BRCA-9	BRCA	71.80	0.41	0.46	27.02	BRCA-10	BRCA	71.50	0.54	0.18	32.85
sample ID	cancer type	coverage	ichorCNA-estimated ctDNA content	median VAF	fraction of cfDNA fragments <150bp (%)																																																														
BRCA-1	BRCA	86.98	0.70	0.11	21.97																																																														
BRCA-2	BRCA	79.41	0.76	0.37	29.35																																																														
BRCA-3	BRCA	94.33	0.13	0.20	27.23																																																														
BRCA-4	BRCA	89.86	0.66	0.25	22.74																																																														
BRCA-5	BRCA	87.92	0.59	0.17	29.61																																																														
BRCA-6	BRCA	85.45	0.51	0.06	24.68																																																														
BRCA-7	BRCA	78.51	0.54	0.32	31.02																																																														
BRCA-8	BRCA	76.67	0.74	0.01	26.47																																																														
BRCA-9	BRCA	71.80	0.41	0.46	27.02																																																														
BRCA-10	BRCA	71.50	0.54	0.18	32.85																																																														
Changes to manuscript	We provide the max VAF values and proportions of short fragments used for the BRCA sample selection in Table S1. Besides, as Supplementary data 1 and 2, we provided BED files of the gene panels used for SNV analysis in																																																																		

	CRC and BRCA, respectively. Furthermore, we have improved the description of the sample selection in the manuscript as shown below. [Page 4 line 15, excerpt] “We performed targeted gene sequencing on colorectal and breast cancer plasma samples and selected samples with high SNV VAFs (indicating high ctDNA burden) for deep ~90x cfDNA WGS (Table S1).” [Methods: Page 23 line 14, excerpt] ”We first used a targeted sequencing panel (Table S7) to screen plasma samples from CRC patients and selected 12 samples (Table S1) of likely high ctDNA burden, having maximum VAF > 15% for known CRC cancer driver mutations (Supplementary Data 1). Similarly, we selected 10 BRCA plasma samples of high ctDNA burden, with either VAF > 15% based on a panel of 77 genes (Table S13) of common breast cancer mutations (Supplementary Data 2), or alternatively, significant proportions (>20%) of short (length < 150bp) cfDNA fragments (Table S1). It has been reported that short cfDNA fragments below 150 bp are enriched in high-ctDNA plasma samples (Mouliere et al., Sci Transl Med, 2018). Deep WGS (~90x) was performed on the 12 cfDNA samples from 7 CRC patients and 10 cfDNA samples from 10 BRCA patients.”
--	---

Reviewer 2, Minor Comment 2: misleading text

Reviewer Comment	In Table S6, please indicate that “coverage” corresponds to the coverage of sWGS data. In the main text, this table refers to “targeted sequencing assay” with a coverage of ~300x. Please correct this.
Author Response	We thank the reviewer for careful reading.
Changes to manuscript	We have corrected “coverage” to “lp-WGS coverage” in the supplementary table and removed “Table S6” in the main text to avoid misunderstanding.

Reviewer 2, Minor Comment 3: correlations between CRC and CRC+BRCA models

Reviewer Comment	Can the authors show correlations between ctDNA levels determined by “CRC-model” and “CRC+BRCA” model in CRC patients? What is the concordance between predicted ctDNA fractions by the two separate models?
Author Response	We thank the reviewer for this good suggestion to show the correlation between ctDNA fractions predicted by the CRC and CRC+BRCA model. As shown in the figure below, the models show high concordance in the predicted ctDNA fractions of the CRC samples in the test set (Pearson $r = 0.95$; Spearman correlation, $\rho = 0.95$).  Fig. S11 Comparison of the ctDNA fractions determined by the CRC model and the “CRC+BRCA” model for the CRC samples in the test set.
Changes to manuscript	We have added the figure above as Fig. S11 in supplementary. Also, we have added this section in the paper as shown below. [Page 18 line 16, excerpt] “We also observed strong concordance with predicted ctDNA fractions from the CRC-specific model when applied to the test set CRC samples (Fig. S11; Pearson $r = 0.95$; Spearman correlation, $\rho = 0.95$).”

REVIEWER COMMENTS

Reviewer #1 (Remarks to the Author):

This was an excellent revision, and addressed effectively all of my concerns. My sole remaining recommendation is with regard to Reviewer #1 Major #3: please also update the Methods and report in the main text the magnitude of those correlations detected for the reader.

Reviewer #2 (Remarks to the Author):

In the revised version of the manuscript “Tissue-specific cell-free DNA degradation quantifies circulating tumor DNA burden” by Zhu et al., the authors have addressed most of my comments which has strengthened the quality of the manuscript. However, there are still concerns that require further clarification. In addition to the main concerns, to avoid any ambiguity I would recommend to improve the clarity in reporting the data in the main text, figures and adding specific details regarding each cohort (number of samples, appropriate referencing to the Methods section, correct naming conventions) included in each analysis. Please find my comments outlined below:

Main concerns:

- The authors have designed a novel capture panel-based assay targeting selected NDRs inferred from WGS data. Despite extending the analysis by assessing the limit of detection in WGS data in the revised version of the manuscript, to benchmark this novel approach the performance needs to be assessed in the actual capture-based data rather than in the original WGS, as each method is subjected to different analytical biases. Considering the fact that the capture process is commonly associated with such technical bias, defining the assay performance by including a cohort of healthy individuals (as suggested previously) is crucial. The assumption that no NDR-based ctDNA detection in cancer patients that had also failed to be detected by alternative methods (SNV- and SCNA-based) can be extrapolated to assess the specificity is not correct. Please provide the relevant data on healthy controls tested by the NDR capture-based assay. Additionally, since this is a customized capture, it would be useful to report the actual target capture design covering the 6 NDR sites.
- Please check the calculations of the relative coverage score as described in the Methods section. To report the differences in relative coverage at NDRs between CRC and healthy individuals, the current score is based on the standard deviation (SD) of CRC patients instead of the SD from a control group (healthy individuals). If the analysis aimed to show the differences based on the z score statistics, this should be corrected (please refer to other publications, e.g. Ulz et al. Nat Commun, 2019).

Additional comments:

- Page 4, line 13: First two sentences are repetitive, please amend this. It should be clear from the text how many samples in each cohort were tested (e.g., currently the overview does not state how many patients were tested by WGS or targeted approach).
- Page 4, line 16: Please indicate directly in the text (by using a reference to Supplementary data and Methods section) which targeted protocol has been used.
- Page 4, line 21: Please add which existing methods do the authors refer to.

- Page 8, line 9: Inconsistency in the number of patients between the text and the figure - the number of CRC patients in the text is 8 while Figure 3a shows 12 patients.
- Page 8, line 21: “ $P < 2.2 \times 10^{-16}$, Wilcoxon rank-sum test” – please indicate in the text that this is presented in Figure 3b.
- Page 10, line 6: Please specify how many patients/healthy individuals were included to generate random subsets of data.
- Page 10, line 11: Which “independent healthy samples” were used in this analysis? Were these samples different from those used for training the model used before, or was it the same cohort that has been just split into two? This is not clear from the text, hence reporting the number of samples for each separate analysis is important.
- Page 13, line 6: Please indicate in the text which samples were included for targeted sequencing (~6000x).
- In the section “Targeted NDR assay to estimate ctDNA burden”, please indicate how many cases of the 53 CRC patients were detected by NDR analysis and compare the detection rates with ichorCNA and SNV-based method to show the benefit of using NDR-based approach.
- Page 18, line 6: Please indicate which 20 tissues were used for building the “blood-specific CRC and BRCA” NDR model and describe this in the Methods section.
- Page 18, line 18: Fig. S11 shows the concordance of ctDNA fractions between CRC and BRCA-CRC models, but this is not clearly explained in the text.

Comments on the figures:

- Figure 3a: CRC model was developed using both tumour- and blood-specific NDRs, yet the figure only shows tumour-specific NDR. Currently, the readers might get an impression that only tumour-specific NDRs were included. Similar issue is shown in Figure 1, with schematics only for tumour-specific NDRs. For completeness, it would be appropriate to report the results (and amend schematics) for both, so it is clear that CRC model used both types of NDRs. The analogous figure (Figure 5a) describing CRC-BRCA model correctly shows only the blood-specific NDRs.
- Figure 3b: The differences between healthy individuals and cancer patients are not shown, while Figure 5a includes this comparison. The legend should be shown as boxes instead of lines.
- Fig. S2: Please name the y axis using the whole name of the “r”
- Fig. S5: The description in the legend is confusing as it might read that this data represents correlations of ctDNA assessed in healthy individuals. Please indicate which cohort of cancer patients was used.
- Figure S6a: The revised version of this figure, highlighting the threshold and by extension the positive and negative signal in the zoomed case CRC-11, should be incorporated into the document. Please highlight the threshold and indicate by colour coding all non-detected samples (as done in the revised Figure S6a). Please amend this in all plots, where appropriate.
- Figure S6b: The labels of axis should refer to ctDNA fractions. Please correct this. The same comment applies to all relevant figures.
- Figure S11: The labels of axis should be “ctDNA fractions by ...”
- Table S5, Table S12: There is a confusion in assigning the axis labels and variables. Please change the variable name “Observed/expected ctDNA fractions” to “Expected cfDNA fractions” - this corresponds to a priori known information based on independent ctDNA measurement, such as SNV-based method and simulation of data. “Predicted ctDNA fraction” should be changed to “Observed ctDNA fraction”, as this was measured (hence observed) by the new approach. Please

refer to other publications (Adalsteinsson et al. Nat Commun, 2017; Newman et al. Nat Biotechnol, 2016) for correct naming conventions.

- Importantly, the x axis should indicate “Expected ctDNA fraction”, while y axis should represent “Observed ctDNA fraction”. This axis assignment should be used throughout the whole manuscript in all relevant figures (main and supplementary), tables and the text.

Minor grammar points:

- Please avoid using description “healthy sample/data” and “cancer sample/data”. Instead please use “data/samples from healthy individuals/cancer patients”. Please correct this.
- Please change “kbp” to “kb”
- “In silico” should be in italics
- “whole-blood cells” should be without hyphen “whole blood cells”
- Figure 3, Figure 5: Spelling error – “preparation” change to “preparation”
- Methods: Change “degC” to “oC”
- Methods: Please change the name “QIAamp circulating nucleic acids kit” and “QiaAmp Circulating Nucleic Acids kit (Qiagen)” to the correct name “QIAamp Circulating Nucleic Acid Kit”
- Please change “N” to “n” to indicate the number of cases
- Page 5, line 3: Please change “cost-effectively monitoring” to “cost-effective monitoring”
- Page 13, line 18: Please change “collected for” to “collected from”

General response to reviewers

We thank the reviewers for supportive feedback and valuable suggestions. Below, we have responded to the specific comments raised.

Reviewer 1 comments	2
Reviewer 1, Comments	2
Reviewer 2 comments	4
Reviewer 2, General Comments	4
Reviewer 2, Major Comment 1: controls by NDR capture-based assay	4
Reviewer 2, Major Comment 2: relative coverage score	6
Reviewer 2, Additional Comment 1: sample clarification	7
Reviewer 2, Additional Comment 2: targeted protocol	7
Reviewer 2, Additional Comment 3: existing methods	8
Reviewer 2, Additional Comment 4: the number of CRC patients	8
Reviewer 2, Additional Comment 5: Wilcoxon rank-sum test	9
Reviewer 2, Additional Comment 6: individuals in random subsets	9
Reviewer 2, Additional Comment 7: independent healthy samples	9
Reviewer 2, Additional Comment 8: samples for targeted sequencing	10
Reviewer 2, Additional Comment 9: detection rates	10
Reviewer 2, Additional Comment 10: 20 tumor tissues	11
Reviewer 2, Additional Comment 11: explanation on Fig. S11	11
Reviewer 2, Comments on Graphics 1: Fig. 3a, Fig. 1	12
Reviewer 2, Comments on Graphics 2: Fig. 3b	13
Reviewer 2, Comments on Graphics 3: Fig. S2	14
Reviewer 2, Comments on Graphics 4: Fig. S5	14
Reviewer 2, Comments on Graphics 5: Figure S6a	15
Reviewer 2, Comments on Graphics 6: Figure S6b	16
Reviewer 2, Comments on Graphics 7: Fig. S11	16
Reviewer 2, Comments on Graphics 8: Table S5, Table S12	17
Reviewer 2, Comments on Graphics 9: expected/observed ctDNA fraction	17
Reviewer 2, Minor Grammar Points	18

Point-by-point response to reviewers

Reviewer 1 comments

Reviewer 1, Comments

Reviewer Comment	This was an excellent revision, and addressed effectively all of my concerns. My sole remaining recommendation is with regard to Reviewer #1 Major #3: please also update the Methods and report in the main text the magnitude of those correlations detected for the reader.
Author Response	We thank the reviewer for the appreciation of our work and for providing great suggestions to improve the manuscript.
Changes to manuscript	We have added the magnitude of the correlations in the main text and updated the Methods accordingly. [Page 6 line 17, excerpt] “To further explore the factors affecting cfDNA degradation at NDRs, we explored the association between NDR relative coverage and a range of epigenetic features (Fig. S2). In addition to gene expression levels (linear regression, promoter $r = -0.23$, junction $r = -0.22$), relative coverage was negatively correlated with DNase hypersensitivity (promoter $r = -0.60$, junction $r = -0.55$), H3K4me3 (promoter $r = -0.59$, junction $r = -0.54$), and H3K27ac (promoter $r = -0.45$, junction $r = -0.41$), which are markers of open chromatin, active promoters, and active enhancers respectively. In contrast, H3K36me3 (promoter $r = 0.49$, junction $r = 0.46$) and H3K9me3 (promoter $r = 0.11$, junction $r = 0.10$), markers of gene bodies and heterochromatin, were positively correlated with NDR relative coverage.” [“Methods”, Page 26 line 5, excerpt] “To explore the association between relative coverage and a range of epigenetic features, we used linear regression to fit each candidate feature (covariate) with relative coverage (response). Whole blood gene expression (fpkm) was discretized into 6 bins [unexpressed, $0.01 < \text{fpkm} \leq 0.1$, $0.1 < \text{fpkm} \leq 1$, $1 < \text{fpkm} \leq 5$, $5 < \text{fpkm} \leq 30$, $\text{fpkm} > 30$] and fitted as a categorical covariate with the unexpressed group as the reference group. Peaks of epigenetic features [DNase, H3K4me3, H3K36me3, H3K27ac, H3K4me1, H3K9me3 and H3K27me3] from primary T-cells (E034) were obtained from the Roadmap Epigenomics Project.

	Epigenetic features were fitted as binary covariates with no signal as the reference group.”
--	--

Reviewer 2 comments

Reviewer 2, General Comments

Reviewer Comment	In the revised version of the manuscript “Tissue-specific cell-free DNA degradation quantifies circulating tumor DNA burden” by Zhu et al., the authors have addressed most of my comments which has strengthened the quality of the manuscript. However, there are still concerns that require further clarification. In addition to the main concerns, to avoid any ambiguity I would recommend to improve the clarity in reporting the data in the main text, figures and adding specific details regarding each cohort (number of samples, appropriate referencing to the Methods section, correct naming conventions) included in each analysis. Please find my comments outlined below:
Author Response	We would like to thank the reviewer for acknowledgment of the improved manuscript and providing helpful suggestions to improve the manuscript. Below we provide a point-by-point reply to the reviewer’s comments.

Reviewer 2, Major Comment 1: controls by NDR capture-based assay

Reviewer Comment	The authors have designed a novel capture panel-based assay targeting selected NDRs inferred from WGS data. Despite extending the analysis by assessing the limit of detection in WGS data in the revised version of the manuscript, to benchmark this novel approach the performance needs to be assessed in the actual capture-based data rather than in the original WGS, as each method is subjected to different analytical biases. Considering the fact that the capture process is commonly associated with such technical bias, defining the assay performance by including a cohort of healthy individuals (as suggested previously) is crucial. The assumption that no NDR-based ctDNA detection in cancer patients that had also failed to be detected by alternative methods (SNV- and SCNA-based) can be extrapolated to assess the specificity is not correct. Please provide the relevant data on healthy controls tested by the NDR capture-based assay. Additionally, since this is a customized capture, it would be useful to report the actual target capture design covering the 6 NDR sites.
Author Response	Firstly, we completely agree with and appreciate the reviewer’s point that the capture process typically has specific technical bias, which might limit the models performance on the targeted sequencing data. However, we would like to emphasize that while the CRC model was indeed trained on WGS data, it robustly predicted ctDNA fractions when applied to targeted sequencing data across independent CRC plasma samples with highly variable ctDNA fractions (Fig. 4b,c, shown below).

Fig. 4 Targeted NDR assay to quantify ctDNA burden and monitor cancer progression. (a) Schematic showing how targeted NDR sequencing, low-pass WGS, and targeted gene sequencing was performed on a cohort of 53 CRC plasma samples. (b) Comparison of ctDNA fractions inferred by targeted NDR-sequencing and low-pass WGS (ichorCNA). (c) Comparison of ctDNA fractions inferred by targeted NDR-sequencing and maximum VAFs (maximum VAF of all SNVs identified in a given plasma sample).

Secondly, while we have tested the model thoroughly using WGS data from healthy individuals (Table S5 and S12, Fig. S5), we agree with the reviewer that it would have been interesting and relevant to also test for potential technical bias in the targeted assay in the healthy samples. However, unfortunately, we don't have additional healthy plasma samples in the laboratory, and due to the COVID-19 pandemic, access to new clinical samples is currently impossible. Instead, we have added a section to better emphasize the limitations of our work, and that the current limit of detection evaluation is based on WGS data, and not the targeted sequencing assay.

Finally, we thank the reviewer for the great suggestion of reporting the target capture design of the 6 NDRs. We have added the target capture design as a new **Suppl. Data 3** in the supplemental material.

Changes to manuscript

We have added the target capture design covering the 6 NDRs as **Suppl. Data 3** in the supplemental material.

We have further commented on the limitations of the current model in the manuscript:

[Page 21 line 14, excerpt]

“However, studies across multiple cancer types are needed to further test and establish the robustness of such an approach. We also recognize that the targeted NDR sequencing approach should be tested across larger patient cohorts and healthy individuals to more accurately assess potential technical bias and its limit of detection.”

Reviewer 2, Major Comment 2: relative coverage score

Reviewer Comment	Please check the calculations of the relative coverage score as described in the Methods section. To report the differences in relative coverage at NDRs between CRC and healthy individuals, the current score is based on the standard deviation (SD) of CRC patients instead of the SD from a control group (healthy individuals). If the analysis aimed to show the differences based on the z score statistics, this should be corrected (please refer to other publications, e.g. Ulz et al. Nat Commun, 2019).
Author Response	We thank the reviewer for highlighting this point. The relative coverage score was used to highlight that CRC-specific genes show greater depletion of cfDNA coverage at NDRs in CRC patients as compared to healthy controls (Fig. 3b). As the reviewer points out, while this score normalizes for the variance in coverage observed in cancer samples, it only considers the mean, and not variance, in the healthy controls. The technical reason for this is that we need sufficient sequencing depth (>50x, Fig. S12, included below) to estimate relative coverage in a given sample. Since the 29 healthy controls only had low-pass WGS (~5x) data, we were only able to estimate mean relative coverage, and not variance (SD), for the cohort of healthy samples.  Fig. S12 Comparison of the observed ctDNA fractions in the 53 original cfDNA samples with capture-based NDR sequencing (mean coverage ~300x) and their downsampled counterparts (100x, 50x, 25x, and 10x, respectively).
Changes to manuscript	We have revised the text in the manuscript to clarify the standard deviation term in the relative coverage score. [Page 25 line 23, excerpt] “s.d.(CRC) is the standard deviation of average relative coverages at NDRs across CRC samples. The variance in healthy samples could not be estimated due to low sequencing depth (~5x).”

Reviewer 2, Additional Comment 1: sample clarification

Reviewer Comment	Page 4, line 13: First two sentences are repetitive, please amend this. It should be clear from the text how many samples in each cohort were tested (e.g., currently the overview does not state how many patients were tested by WGS or targeted approach).
Author Response	We thank the reviewer for pointing this out. We have deleted the repetitive sentence to provide a clear description of the cohort. The cohorts and samples/patients are further described in the Methods section and in supplementary tables (Table S1, S6).
Changes to manuscript	We have deleted the repetitive sentence and provided a clearer description of the cohort and samples used. [Page 4 line 13, excerpt] “We generated deep cfDNA WGS profiles of healthy individuals as well as cancer patients with high ctDNA burden (Fig. 1). We collected blood samples (n=29) from healthy individuals and extracted plasma cfDNA for paired-end WGS (merged ~150x coverage) (Fig. 1). We performed targeted sequencing (see Methods) of plasma samples (CRC n=65, BRCA n=36) from cancer patients and selected samples (CRC n=12, BRCA n=10) with high SNV VAFs (indicating high ctDNA burden) for deep ~90x cfDNA WGS (Table S1).....” we designed a compact (<25 kb) capture-based sequencing assay targeting predictive NDRs to explore the robustness of NDR-based targeted approach using independent plasma samples (n=53) from CRC patients.....”

Reviewer 2, Additional Comment 2: targeted protocol

Reviewer Comment	Page 4, line 16: Please indicate directly in the text (by using a reference to Supplementary data and Methods section) which targeted protocol has been used.
Author Response	We thank the reviewer for this suggestion. We added “see Methods” in the text to indicate the targeted protocol is described in detail in the Methods section.
Changes to manuscript	We have revised the text in the manuscript.

	[Page 4 line 15, excerpt] “We performed targeted sequencing (see Methods) of plasma samples (CRC n=65, BRCA n=36) from cancer patients”
--	--

Reviewer 2, Additional Comment 3: existing methods

Reviewer Comment	Page 4, line 21: Please add which existing methods do the authors refer to.
Author Response	In line with the reviewer’s suggestion, we have added the references to indicate which four methods we used to infer ctDNA fractions. We also note that these four methods are further introduced in the Results and Methods sections.
Changes to manuscript	We have added the references of the methods in the text. [Page 4 line 18, excerpt] “In these high ctDNA burden WGS samples, we could obtain ctDNA burden estimates using existing methods (Bao et al., 2014; Ha et al., 2014; Larson and Fridley, 2013; Oesper et al., 2014) (see Methods) that infer tumor purity using matched tumor and germline high-depth WGS data.”

Reviewer 2, Additional Comment 4: the number of CRC patients

Reviewer Comment	Page 8, line 9: Inconsistency in the number of patients between the text and the figure - the number of CRC patients in the text is 8 while Figure 3a shows 12 patients.
Author Response	We thank the reviewer for careful reading and apologise for the confusion caused by our typo. The number should be “8”, and we have corrected it in Fig. 3a.
Changes to manuscript	We have corrected the number to “n=8” in Fig. 3a.

Reviewer 2, Additional Comment 5: Wilcoxon rank-sum test

Reviewer Comment	Page 8, line 21: “$P < 2.2 \times 10^{-16}$, Wilcoxon rank-sum test” – please indicate in the text that this is presented in Figure 3b.
Author Response	We have added “Fig. 3b” in the text to indicate the “Wilcoxon rank-sum test” is presented in Fig. 3b. Thank you.
Changes to manuscript	We have added “Fig. 3b” in the text accordingly. [Page 8 line 14, excerpt] “Furthermore, directly comparing CRC and blood-specific genes, CRC-specific genes had significantly greater cfDNA depletion at NDRs in plasma samples from CRC patients ($P < 2.2 \times 10^{-16}$, Wilcoxon rank-sum test, Fig. 3b).”

Reviewer 2, Additional Comment 6: individuals in random subsets

Reviewer Comment	Page 10, line 6: Please specify how many patients/healthy individuals were included to generate random subsets of data.
Author Response	We thank the reviewer for this comment. We have added the information on the number of plasma samples from patients and healthy individuals that were included to generate random subsets of data.
Changes to manuscript	We have revised the text accordingly. [Page 9 line 23, excerpt] “We used the 113 in silico test set CRC samples (Fig. 3e, CRC-9 to 12) as positives, and 40 random subsets (Table S5, ~80x each) from the data of plasma samples ($n=29$) from healthy individuals as negatives.”

Reviewer 2, Additional Comment 7: independent healthy samples

Reviewer Comment	Page 10, line 11: Which “independent healthy samples” were used in this analysis? Were these samples different from those used for training the model used before, or was it the same cohort that has been just split into two? This is not clear from the
------------------	---

	text, hence reporting the number of samples for each separate analysis is important.
Author Response	We agree that using ‘independent’ in this context could be confusing. The healthy samples were split into two groups (training and testing), and the withheld testing group was referred to as an ‘independent’ set not seen during model training. To avoid confusion, we have revised and clarified this further in the text.
Changes to manuscript	We have amended the text accordingly. [Page 10 line 6, excerpt] “To further evaluate the robustness of the model when tested on in silico samples generated using healthy samples not seen during model training, we split the healthy samples (n=29) into two different groups to separately generate in silico training and test data.”

Reviewer 2, Additional Comment 8: samples for targeted sequencing

Reviewer Comment	Page 13, line 6: Please indicate in the text which samples were included for targeted sequencing (~6000x).
Author Response	We thank the reviewer for this suggestion. We have added the number of samples and referenced “(Fig. 4a, Table S6)” in the text to indicate the details of the samples included for the targeted sequencing.
Changes to manuscript	We have revised the text to indicate which samples were included for the targeted sequencing. [Page 13 line 12, excerpt] “Moreover, in the 53 samples (Fig. 4a, Table S6), we performed targeted sequencing (~6000x) of a panel of 100 frequently mutated genes (~370 kb, Table S7) in colorectal cancer.”

Reviewer 2, Additional Comment 9: detection rates

Reviewer Comment	In the section “Targeted NDR assay to estimate ctDNA burden”, please indicate how many cases of the 53 CRC patients were detected by NDR analysis and compare the detection rates with ichorCNA and SNV-based method to show the benefit of using NDR-based approach.
------------------	--

Author Response	We appreciate the reviewer's suggestion and have modified the manuscript accordingly.
Changes to manuscript	We have added a section to highlight the detection rates of the different methods. [Page 13 line 19, excerpt] "ctDNA was detected in 49 out of 53 (92%) samples with the targeted NDR approach, compared to 33/53 (62%) and 27/53 (51%) with ichorCNA and SNV calling approaches, respectively. The 4 ctDNA-negative samples identified with the NDR approach were also ctDNA-negative using ichorCNA and the SNV approach (Table S6)."

Reviewer 2, Additional Comment 10: 20 tumor tissues

Reviewer Comment	Page 18, line 6: Please indicate which 20 tissues were used for building the "blood-specific CRC and BRCA" NDR model and describe this in the Methods section.
Author Response	In line with the reviewer's suggestion, we have added the names of the 20 tissues (cancer types) in the Methods section.
Changes to manuscript	We have added the names of the 20 tumor tissues in the manuscript. [Page 28 line 4, excerpt] "For the CRC+BRCA model, we only shortlisted transcripts with blood-specific expression ($\text{fpkm}_{\text{blood}} > 5$) that were also lowly expressed ($\text{fpkm} < 1$) in tumors of all 20 tumor types (TCGA tumor type acronyms: BLCA, BRCA, CESC, CRC, ESCA, GBM, HNSC, KIRC, KIRP, LGG, LIHC, LUAD, LUSC, OV, PAAD, PRAD, SKCM, STAD, THCA, UCEC), leading to a total of 792 features."

Reviewer 2, Additional Comment 11: explanation on Fig. S11

Reviewer Comment	Page 18, line 18: Fig. S11 shows the concordance of ctDNA fractions between CRC and BRCA-CRC models, but this is not clearly explained in the text.
Author Response	We thank the reviewer for thoroughly reviewing our manuscript and supplementary data. We have revised the text to address this issue.

Changes to manuscript	We have amended the text to explain the concordance of ctDNA fractions between the CRC and BRCA+CRC models. [Page 17 line 23, excerpt] “We also observed strong concordance between the CRC+BRCA and CRC-specific models in their predicted ctDNA fractions in the test set plasma samples from CRC patients (Pearson $r = 0.95$; Spearman correlation, $\rho = 0.95$; Fig. S11).”
--

Reviewer 2, Comments on Graphics 1: Fig. 3a, Fig. 1

Reviewer Comment	Figure 3a: CRC model was developed using both tumour- and blood-specific NDRs, yet the figure only shows tumour-specific NDR. Currently, the readers might get an impression that only tumour-specific NDRs were included. Similar issue is shown in Figure 1, with schematics only for tumour-specific NDRs. For completeness, it would be appropriate to report the results (and amend schematics) for both, so it is clear that CRC model used both types of NDRs. The analogous figure (Figure 5a) describing CRC-BRCA model correctly shows only the blood-specific NDRs.
Author Response	We thank the reviewer for raising this point. We have amended Fig. 3a to show both tumor-specific and blood-specific NDRs that reflect tissue-specific cfDNA relative coverage. Also, we have amended Fig. 1, adding bar plots for both tumor and blood-specific genes/NDRs.
Changes to manuscript	We have updated Fig. 3a, Fig. 1, and the main text accordingly. [Page 8 line 7, excerpt] “As an example, we identified PPP1R16A as a CRC-specific gene with robust depletion of NDR cfDNA coverage in plasma samples from cancer patients as compared to healthy individuals, and GMFG as a blood-specific gene with greater coverage depletion in healthy blood plasma (Fig. 3a).”   Fig. 3 Quantitative estimation of colorectal cancer ctDNA burden. (a) cfDNA relative coverage for the promoter region of PPP1R16A (ENST00000528430) overexpressed

in CRC tumors relative to whole blood, and cfDNA relative coverage for the junction region of *GMFG* (ENST00000602185) overexpressed in whole blood relative to CRC tumors. The dark red curve shows the mean coverage across CRC samples.

Fig. 1 Overview of approach ...

Reviewer 2, Comments on Graphics 2: Fig. 3b

Reviewer Comment	Figure 3b: The differences between healthy individuals and cancer patients are not shown, while Figure 5a includes this comparison. The legend should be shown as boxes instead of lines.
Author Response	We appreciate this suggestion. We have added graphics in Fig. 3a to directly compare cfDNA coverages at NDRs between healthy individuals and cancer patients, similarly as what we showed in Fig. 5a. We also changed the legend to boxes from lines in Fig. 3b.
Changes to manuscript	We have updated Fig. 3a and b accordingly, shown below.

	Fig. 3 Quantitative estimation of colorectal cancer ctDNA burden.....
--	---

Reviewer 2, Comments on Graphics 3: Fig. S2

Reviewer Comment	Fig. S2: Please name the y axis using the whole name of the "r"
Author Response	We thank the reviewer for this comment.
Changes to manuscript	We have amended the name of the y axis in Fig. S2 and also explained the y axis in the legend.     Fig. S2 Correlation between relative coverage of NDRs and epigenetic features. For each candidate covariate/predictor, a linear regression is fitted with relative coverage as the response. The Pearson correlation coefficient (y axis, signed square root of R-squared from regression) is shown for each candidate variable.....

Reviewer 2, Comments on Graphics 4: Fig. S5

Reviewer Comment	Fig. S5: The description in the legend is confusing as it might read that this data represents correlations of ctDNA assessed in healthy individuals. Please indicate which cohort of cancer patients was used.
Author Response	We thank the reviewer for pointing this out. We have amended the legend to avoid the confusion. We have also indicated the cancer cohort used in this analysis in the legend.
Changes to	We have revised the legend of Fig. S5 to clarify this issue.

manuscript

Fig. S5. Model performance on 10 test sets generated using different (withheld) healthy samples from the training sets. Individual normal samples (n=29) in the healthy cohort were evenly split into 2 sets, used to dilute the plasma samples from CRC patients in training (CRC-1 to 8 in Table S1) and test (CRC-9 to 12) sets separately. (a) The correlation (Pearson and Spearman) between the expected and observed ctDNA fractions across the 10 test sets. (b) The mean absolute error (MAE) between the expected and observed ctDNA fractions for the 10 test sets.

Reviewer 2, Comments on Graphics 5: Figure S6a

Reviewer Comment	Figure S6a: The revised version of this figure, highlighting the threshold and by extension the positive and negative signal in the zoomed case CRC-11, should be incorporated into the document. Please highlight the threshold and indicate by colour coding all non-detected samples (as done in the revised Figure S6a). Please amend this in all plots, where appropriate.
Author Response	We thank the reviewer for this great suggestion. Fig. S6 is meant to show the general concordance between the expected and ichorCNA-estimated ctDNA fractions in the in silico samples. Fig. S7 shows a zoomed-in version of the low-ctDNA samples. We have amended Fig. S7 to highlight (in red color) the samples where ctDNA was undetected by ichorCNA.
Changes to manuscript	We have amended Fig. S7 as shown below. Figure S7 is a scatter plot showing 'ctDNA fractions' on the y-axis (ranging from 0 to 0.05) for 12 CRC samples (CRC-1 to CRC-12) on the x-axis. The plot compares 'expected' values (represented by open circles) and 'ichorCNA' estimates (represented by blue crosses). A horizontal dashed line at y=0 represents the detection threshold. Red crosses at the bottom of the plot indicate samples where ctDNA was not detected by ichorCNA. The legend indicates 'low ctDNA samples (≤0.05)' and identifies the symbols for 'expected' and 'ichorCNA'.

	Fig. S7 Performance of ichorCNA when applied to the samples with low ctDNA burden. 31 out of 120 low-ctDNA samples of CRC were predicted as non-cancerous by ichorCNA, highlighted in red. Grey dashed line indicates ctDNA fraction of 0.
--	--

Reviewer 2, Comments on Graphics 6: Figure S6b

Reviewer Comment	Figure S6b: The labels of axis should refer to ctDNA fractions. Please correct this. The same comment applies to all relevant figures.
Author Response	We have corrected the labels of the axis. Thank you.
Changes to manuscript	We have updated Fig. S6 as shown below.  Fig. S6 Comparison of expected and ichorCNA-predicted ctDNA fractions across the CRC cfDNA samples. (a) ctDNA fractions across the CRC cfDNA samples. (b) Comparison of expected and ichorCNA-predicted ctDNA fractions.

Reviewer 2, Comments on Graphics 7: Fig. S11

Reviewer Comment	Figure S11: The labels of axis should be “ctDNA fractions by ...”
Author Response	Thank you for this suggestion, which we have corrected the labels.
Changes to manuscript	We have corrected the labels in Fig. S11, as shown below.

Reviewer 2, Comments on Graphics 8: Table S5, Table S12

Reviewer Comment	Table S5, Table S12: There is a confusion in assigning the axis labels and variables. Please change the variable name “Observed/expected ctDNA fractions” to “Expected cfDNA fractions” - this corresponds to a priori known information based on independent ctDNA measurement, such as SNV-based method and simulation of data. “Predicted ctDNA fraction” should be changed to “Observed ctDNA fraction”, as this was measured (hence observed) by the new approach. Please refer to other publications (Adalsteinsson et al. Nat Commun, 2017; Newman et al. Nat Biotechnol, 2016) for correct naming conventions.
Author Response	This is an excellent recommendation, and we thank the reviewer for spotting this. We have changed the variable names to “Expected cfDNA fraction” (for priori known information) and “Observed ctDNA fraction” (for prediction by method) respectively.
Changes to manuscript	We have updated to use “expected cfDNA fraction” and “observed ctDNA fraction” throughout the whole manuscript including text, figures, and tables.

Reviewer 2, Comments on Graphics 9: expected/observed ctDNA fraction

Reviewer Comment	Importantly, the x axis should indicate “Expected ctDNA fraction”, while y axis should represent “Observed ctDNA fraction”. This axis assignment should be used throughout the whole manuscript in all relevant figures (main and supplementary), tables and the text.
Author	Again, we want to thank the reviewer for this helpful comment.

Response	
Changes to manuscript	We have changed to use “expected ctDNA fraction” and “observed ctDNA fraction” throughout the whole manuscript including the main text, all figures and tables.

Reviewer 2, Minor Grammar Points

Reviewer Comment	 - Please avoid using description “healthy sample/data” and “cancer sample/data”. Instead please use “data/samples from healthy individuals/cancer patients”. Please correct this. - Please change “kbp” to “kb” - “In silico” should be in italics - “whole-blood cells” should be without hyphen “whole blood cells” - Figure 3, Figure 5: Spelling error – “preparation” change to “preparation” - Methods: Change “degC” to “oC” - Methods: Please change the name “QIAamp circulating nucleic acids kit” and “QiaAmp Circulating Nucleic Acids kit (Qiagen)” to the correct name “QIAamp Circulating Nucleic Acid Kit” - Please change “N” to “n” to indicate the number of cases - Page 5, line 3: Please change “cost-effectively monitoring” to “cost-effective monitoring” - Page 13, line 18: Please change “collected for” to “collected from”
Author Response	We thank the reviewer for careful reading and kind help on the grammar checking. We have corrected all instances of the grammar errors in the paper.
Changes to manuscript	We have corrected all the grammar errors above in the main text and figures in the manuscript.

REVIEWER COMMENTS

Reviewer #2 (Remarks to the Author):

I appreciate the efforts of the authors to address all additional points that I have raised in the previous revision. The quality of the manuscript has been substantially improved. The study is original, conceptually interesting and would undoubtedly bring an advancement in the field. Unfortunately, the two main drawbacks pointed out previously have not been addressed appropriately.

1) the lack of plasma samples from healthy individuals analyzed by targeted NDR assay: this approach represents the actual potential clinical application for a low-cost monitoring of ctDNA and therefore it needs appropriate assessment available for the scientific community. It is appreciated that the CRC model applied to targeted sequencing data correctly predicted ctDNA fractions across independent CRC plasma samples. However, this observation does not relate to the test specificity which can be only assessed by performing a targeted capture in a cohort of healthy controls. The authors mentioned that due to the current pandemic situation they do not have the access to clinical samples. As a solution, the plasma samples from healthy individuals can be obtained from commercially available sources that are commonly used for analytical assessments of cfDNA assays.

2) the second concern is the calculation of the relative coverage score (using a z-score statistics), which should consider the SD of healthy controls. The explanation provided in the rebuttal letter is not clear and only points out the insufficient sequencing depth of controls. The goal of this analysis should be to assess how many SDs is the value of a tested sample away from the mean of the reference group (plasma of healthy individuals). The samples from cancer patients should not be used as a reference group for determining SD. The need for establishing a group of reference samples have been previously outlined by Ulz et al. Nat Commun (2019) who have used a similar approach for assessment of the coverage across TFBS. Here, "the z-scores were calculated for every transcription factor from the accessibility ranks by taking the respective rank and subtracting the mean rank of the control samples and dividing by the standard deviation of the transcription factor ranks of the control samples".

Reviewer 2 comments

Reviewer 2, General Comments

Reviewer Comment	I appreciate the efforts of the authors to address all additional points that I have raised in the previous revision. The quality of the manuscript has been substantially improved. The study is original, conceptually interesting and would undoubtedly bring an advancement in the field. Unfortunately, the two main drawbacks pointed out previously have not been addressed appropriately.
Author Response	We would like to thank the reviewer for the encouraging feedback and for providing very helpful suggestions to further improve the quality of the manuscript. Below we provide a point-by-point reply to the reviewer's comments.

Reviewer 2, Comment 1: controls by targeted NDR assay

Reviewer Comment	1) the lack of plasma samples from healthy individuals analyzed by targeted NDR assay: this approach represents the actual potential clinical application for a low-cost monitoring of ctDNA and therefore it needs appropriate assessment available for the scientific community. It is appreciated that the CRC model applied to targeted sequencing data correctly predicted ctDNA fractions across independent CRC plasma samples. However, this observation does not relate to the test specificity which can be only assessed by performing a targeted capture in a cohort of healthy controls. The authors mentioned that due to the current pandemic situation they do not have the access to clinical samples. As a solution, the plasma samples from healthy individuals can be obtained from commercially available sources that are commonly used for analytical assessments of cfDNA assays.
Author Response	We appreciate the reviewer's feedback on this important aspect. As the reviewer points out, we have shown that targeted NDR sequencing can quantify ctDNA levels across 53 independent CRC samples. However, we fully acknowledge that the current sensitivity/specificity for the detection of cancer in healthy individuals is estimated using WGS data, and not using a targeted panel. This is stated as a limitation and consideration for future work in our discussion: "However, studies across multiple cancer types are needed to further test and establish the robustness of such an approach. We also recognize that the targeted NDR sequencing approach should be tested across larger patient cohorts and healthy individuals to more accurately assess potential technical bias and its limit of detection."

	To further address the reviewer’s comment, and to provide full transparency to the scientific community, we have created an accompanying repository for the code/software and validation/benchmarking data on GitHub (https://github.com/skandlab/NDRquant). This repository will hold all code, model coefficients (Table S4), and links to raw data on EGA (EGAS00001004657). Furthermore, we will keep this resource up-to-date with benchmark results across new cohorts, cancer types, and healthy samples as data become available.
Changes to manuscript	We have added a link to the new Github repository: [Page 28 line 7, excerpt] “The NDR models, code, and data accessions are available at https://github.com/skandlab/NDRquant .”

Reviewer 2, Comment 2: relative coverage score

Reviewer Comment	2) the second concern is the calculation of the relative coverage score (using a z-score statistics), which should consider the SD of healthy controls. The explanation provided in the rebuttal letter is not clear and only points out the insufficient sequencing depth of controls. The goal of this analysis should be to assess how many SDs is the value of a tested sample away from the mean of the reference group (plasma of healthy individuals). The samples from cancer patients should not be used as a reference group for determining SD. The need for establishing a group of reference samples have been previously outlined by Ulz et al. Nat Commun (2019) who have used a similar approach for assessment of the coverage across TFBS. Here, “the z-scores were calculated for every transcription factor from the accessibility ranks by taking the respective rank and subtracting the mean rank of the control samples and dividing by the standard deviation of the transcription factor ranks of the control samples”.
Author Response	We thank the reviewer for this comment and apologize for the lack of clarity in our previous response. The 29 healthy controls each only had lpWGS (~5x) data and thus lacked sufficient sequencing depth (>50x required, Fig. S12) to measure the relative coverage (and standard deviation) at individual NDR sites. Instead, to account for variability across samples in our Relative Coverage score, we chose to estimate the variance from the high depth (~90x) cancer WGS samples. As a note, Ulz et al. considered pooled TFBSs, with >1000 TFBS regions per TF, and could therefore estimate the standard deviation across the aggregate/collection of all these sites with lpWGS data from individual samples. We do acknowledge the reviewer’s point that variance estimated from plasma samples of cancer patients could differ from samples of healthy individuals.

Therefore, to further address the reviewer's comment, we used a **bootstrapping approach to estimate the variance in healthy individuals**. Briefly, we generated 20 healthy subsets of ~50x coverage, where each subset was generated by randomly selecting (with replacement) and merging 10 samples from the pool of 29 healthy controls. We then re-calculated the relative coverage score using the standard deviation across these 20 healthy control subsets. Consistent with our previous analysis (Fig. 3b), this new analysis confirmed robust cfDNA depletion at promoter/junction NDRs in CRC-specific genes in the plasma of CRC patients as compared to healthy controls (Fig. S14, shown below).

Fig. S14 Relative coverage scores based on the variance of 20 healthy control subsets for the transcripts differentially expressed between CRC tumors and whole blood.

Changes to manuscript

We have added the figure above as Fig. S14 in the supplementary and revised the text in the manuscript accordingly:

[Page 23 line 22, excerpt]

“The variance across individual sites in healthy samples could not be estimated due to low sequencing depth (~5x). Instead, to test for differences in variance between cancer and healthy samples, we approximated the variance in healthy samples using bootstrapping. Briefly, we estimated the standard deviation across 20 subsets of healthy samples (~50x merged) generated by randomly sampling (with replacement) and merging 10 samples out of 29 healthy controls. This analysis showed similar separation between CRC and blood-specific NDRs when variance was estimated in plasma samples of cancer (Fig. 3b) and healthy control (Fig. S14).”

REVIEWERS' COMMENTS

Reviewer #2 (Remarks to the Author):

I would like to thank the authors for additional clarification and the analysis in response to my comments. This revision together with the additional comment from the editorial board have addressed all my concerns and I believe that the readers of Nature Communications will find this study very interesting.

Reviewer 2 comments

Reviewer 2, General Comments

Reviewer Comment	I would like to thank the authors for additional clarification and the analysis in response to my comments. This revision together with the additional comment from the editorial board have addressed all my concerns and I believe that the readers of Nature Communications will find this study very interesting.
Author Response	We are very appreciative of the reviewer's constructive comments which has greatly enriched the manuscript.